# Learning Linear Polytree Structural Equation Model

**Xingmei Lou**                                                                *xmlou@ucdavis.edu*
*Department of Statistics*
*University of California, Davis*
*Davis, CA 95616, USA*

**Yu Hu**                                                                         *mahy@ust.hk*
*Department of Mathematics and Division of Life Science*
*The Hong Kong University of Science and Technology*
*Clear Water Bay, Hong Kong S.A.R.*

**Xiaodong Li**                                                               *xdgli@ucdavis.edu*
*Department of Statistics*
*University of California, Davis*
*Davis, CA 95616, USA*

**Reviewed on OpenReview:** *https://openreview.net/forum?id=N28FdYO2sH*

## Abstract

We are interested in the problem of learning the directed acyclic graph (DAG) when data are generated from a linear structural equation model (SEM) and the causal structure can be characterized by a polytree. Under the Gaussian polytree models, we study sufficient conditions on the sample sizes for the well-known Chow-Liu algorithm to exactly recover both the skeleton and the equivalence class of the polytree, which is uniquely represented by a CPDAG. On the other hand, necessary conditions on the required sample sizes for both skeleton and CPDAG recovery are also derived in terms of information-theoretic lower bounds, which match the respective sufficient conditions and thereby give a sharp characterization of the difficulty of these tasks. We also consider the problem of inverse correlation matrix estimation under the linear polytree models, and establish the estimation error bound in terms of the dimension and the total number of v-structures. We also consider an extension of group linear polytree models, in which each node represents a group of variables. Our theoretical findings are illustrated by comprehensive numerical simulations, and experiments on benchmark data also demonstrate the robustness of polytree learning when the true graphical structures can only be approximated by polytrees.

## 1  Introduction

Over the past three decades, the problem of learning directed graphical models from i.i.d. observations of a multivariate distribution has received an enormous amount of attention since they provide a compact and flexible way to represent the joint distribution of the data, especially when the associated graph is a directed acyclic graph (DAG), which is a directed graph with no directed cycles. DAG models are popular in practice with applications in biology, genetics, machine learning, and causal inference (Sachs et al., 2005; Zhang et al., 2013; Koller & Friedman, 2009; Spirtes et al., 2000). There exists extensive literature on learning the graph structure from i.i.d. observations under DAG models. For a summary, see the survey papers by Drton & Maathuis (2017) and Heinze-Deml et al. (2018). Existing approaches generally fall into two categories, constraint-based methods (Spirtes et al., 2000; Pearl, 2009) and score-based methods (Chickering, 2002b). Constraint-based methods utilize conditional independence tests to determine whether there exists an edge between two nodes and then orient the edges in the graph, such that the resulting graph is compatible

with the conditional independencies determined in the data. Score-based methods formulate the structure learning task as optimizing a score function based on the unknown graph and the data.

A polytree is a connected DAG that contains no cycles even if the directions of all edges are ignored. It is practically useful due to tractability in both structure learning and inference. To the best of our knowledge, structure learning of polytree models was originally studied in Rebane & Pearl (1987), in which the skeleton of the polytree is estimated by applying the Chow-Liu algorithm (Chow & Liu, 1968) to pairwise mutual information quantities, a method that has been widely used in the literature of Markov random field to fit undirected tree models. Polytree graphical models have received a significant amount of research interest both empirically and theoretically ever since, see, e.g., Dasgupta (1999); Cheng et al. (2002), and recent efforts such as Chatterjee & Vidyasagar (2022); Tramontano et al. (2022).

This paper aims to study sample size conditions of the method essentially proposed in Rebane & Pearl (1987) for the recovery of polytree structures by applying the Chow-Liu algorithm to pairwise sample correlations in the case of Gaussian linear structure equation models (SEM). We establish sufficient conditions on the sample sizes for consistent recovery of both the skeleton and equivalence class for the underlying polytree structure. On the other hand, we will also establish the necessary conditions on the sample sizes for these two tasks through information-theoretic lower bounds. Our sufficient and necessary conditions match in order in a broad regime of model parameters, and thereby characterize the difficulty of these two tasks in polytree learning.

A relevant line of research is structure learning for tree-structured undirected graphical models, including both discrete cases (Heinemann & Globerson, 2014; Bresler & Karzand, 2020; Netrapalli et al., 2010; Anand-kumar et al., 2012b;a) and Gaussian cases (Tan et al., 2010; Tavassolipour et al., 2018; Nikolakakis et al., 2019; Katiyar et al., 2019). In particular, conditions on the sample size for undirected tree structure learning via the Chow-Liu algorithm have been studied for both Ising and Gaussian models (Bresler & Karzand, 2020; Tavassolipour et al., 2018; Nikolakakis et al., 2019), and the analyses usually rely crucially on the so-called "correlation decay" property over the true undirected tree. The correlation decay properties can usually be explicitly quantified by the pairwise population correlations corresponding to the edges of the underlying true tree. Based on this result and some perturbation results of pairwise sample correlations to their population counterparts, sufficient conditions on the sample size for undirected tree recovery with the Chow-Liu algorithm can be straightforwardly obtained.

In order to apply the above technical framework to study the sample size conditions for polytree learning, a natural question is whether we have a similar correlation decay phenomenon for the polytree models. In fact, this is suggested in the seminal paper Rebane & Pearl (1987). To be concrete, under some non-degeneracy assumptions, it has been shown in Rebane & Pearl (1987) (see their Eq. 13) that there holds a "mutual information decay" along the skeleton of the polytree. In broad terms, the mutual information decay is a direct implication of the well-known "data processing inequality" in information theory (Thomas & Joy, 2006). Restricted to the very special case of Gaussian linear SEM, the mutual information decay is indeed equivalent to the property of population correlation decay.

To obtain some meaningful sample complexity result, we need to quantify such correlation decay explicitly as what has been done in the study of the Chow-Liu algorithm for undirected tree models (Bresler & Karzand, 2020; Tavassolipour et al., 2018; Nikolakakis et al., 2019). The mutual information decay given in Rebane & Pearl (1987) holds for general polytree models, but one can expect to further quantify such decay under more specific models. In fact, if we restrict the polytree model to linear SEM, by applying the well-known Wright's formula (Wright, 1960; Nowzohour et al., 2017; Foygel et al., 2012), the population correlation decay property can be quantified by the pairwise correlations corresponding to the tree edges. With such quantification of correlation decay over the underlying polytree skeleton, we can apply the ideas from undirected tree structure learning to establish sufficient conditions on sample size for polytree skeleton recovery via the Chow-Liu algorithm. In broad terms, if the maximum absolute correlation coefficient over the polytree skeleton is uniformly bounded below 1, the Chow-Liu algorithm recovers the skeleton exactly with high probability if the sample size satisfies $n > O((\log p)/\rho_{\min}^2)$, where $p$ is the number of variables and $\rho_{\min}$ is the minimum absolute population correlation coefficient over the skeleton.

To determine the directions of the polytree over the skeleton, the concept of CPDAG (Verma & Pearl, 1991) captures the equivalence class of polytrees. We then consider the CPDAG recovery procedure introduced in Verma & Pearl (1992) and Meek (1995), which is a polynomial time algorithm based on identifying all the v-structures (Verma & Pearl, 1991). Therefore, conditional on the exact recovery of the skeleton, recovering the CPDAG is equivalent to recovering all v-structures. In a non-degenerate polytree model, a pair of adjacent edges form a v-structure if and only if the two non-adjacent node variables in this triplet are independent, so we consider a natural v-structure identification procedure by thresholding the pairwise sample correlations over all adjacent pairs of edges with some appropriate threshold. In analogy to the result of skeleton recovery, we show that the CPDAG of the polytree can be exactly recovered with high probability if the sample size satisfies $n > O((\log p)/\rho_{\min}^4)$. Furthermore, by using Fano's method, we show that $n > O((\log p)/\rho_{\min}^2)$ is necessary for skeleton recovery, while $n > O((\log p)/\rho_{\min}^4)$ is necessary for CPDAG recovery. This means that we have sharply characterized the difficulties for the two tasks. We briefly note studies on linear SEMs that do not assume a polytree structure, such as Peters & Bühlmann (2014); Ghoshal & Honorio (2018). In these works, the authors make alternative assumptions to ensure the identifiability of the DAG, for instance, by assuming equal noise variances. For a discussion of the general SEM literature, we refer the readers to Ghoshal & Honorio (2018).

The paper is organized as follows: In Section 2, we review the concepts of linear polytree SEM, Markov equivalence and CPDAG, and the polytree learning method based on the Chow-Liu algorithm. In Section 3, we give optimal sample size conditions for both the skeleton and CPDAG recovery, particularly in terms of the minimum correlation over the tree skeleton. In Section 4, we introduce a version of PC algorithm adapted to the linear polytree models, and establish the same sample size conditions. In Section 5, we discuss a method of estimating the inverse correlation matrix for linear polytree models, and establish an upper bound of estimation in the entry-wise $\ell_1$ norm. Our theoretical findings are empirically demonstrated in Section 7, along with numerical results under some benchmark simulated data in the literature of DAG learning. A brief summary of our work and some potential future research are discussed in Section 8.

## 2 Linear Polytree Models and Learning

This section aims to give an overview of the concepts of linear polytree SEM, equivalence classes characterized by CPDAG, and the Chow-Liu algorithm for polytree learning. Most materials are not new, but we give a self-contained introduction of these important concepts and methods so that our main results introduced in the subsequent sections will be more accessible to a wider audience.

### 2.1 Linear Polytree Models

Let $G = (V, E)$ be a directed graph with vertex set $V = \{1, 2, ..., p\}$ and edge set $E$. We use $i \to j \in E$ to denote that there is a directed edge from node $i$ to node $j$ in $G$. A directed graph with no directed cycles is referred to as a directed acyclic graph (DAG). The parent set of node $j$ in $G$ is denoted as $Pa(j) \coloneqq \{i \in V : i \to j \in E\}$. Correspondingly, denote by $Ch(j) \coloneqq \{k : j \to k \in E\}$ the children set of $j$.

Let $\boldsymbol{x} = [X_1, \ldots, X_p]^\top$ be a random vector where each random variable $X_j$ corresponds to a node $j \in V$. The edge set $E$ usually encodes the causal relationships among the variables. The random vector $\boldsymbol{x}$ is said to be Markov on a DAG $G$ if its joint density function (or mass function) $p(\boldsymbol{x})$ can be factorized according to $G$ as $p(\boldsymbol{x}) = \prod_{j=1}^p p(x_j | x_{Pa(j)})$, where $p(x_j | x_{Pa(j)})$ is the conditional density/probability of $X_j$ given its parents $X_{Pa(j)} \coloneqq \{X_i : i \in Pa(j)\}$. We usually refer to $(G, p(\boldsymbol{x}))$ as a DAG model.

Throughout this work, we restrict our discussion to an important sub-class of DAG models: linear structure equation models (SEM), in which the dependence of each $X_j$ on its parents is linear with additive noise. The parameterization of the linear SEM with directed graph $G = (V, E)$ would be

$$X_j = \sum_{i=1}^p \beta_{ij} X_i + \epsilon_j = \sum_{i \in Pa(j)} \beta_{ij} X_i + \epsilon_j, \quad \text{for } j = 1, \ldots, p, \tag{1}$$

where

$$\beta_{ij} \neq 0 \quad \text{if and only if} \quad i \to j \in E,$$

and all $\epsilon_j$'s are independent with zero mean. Let $\boldsymbol{B} = \begin{bmatrix} \beta_{ij} \end{bmatrix} \in \mathbb{R}^{p \times p}$ and $\boldsymbol{\epsilon} = [\epsilon_1, \ldots, \epsilon_p]^\top$. Then the SEM can be represented as

$$\boldsymbol{x} = \boldsymbol{B}^\top \boldsymbol{x} + \boldsymbol{\epsilon}. \tag{2}$$

Denote $\mathrm{Cov}(\boldsymbol{x}) = \boldsymbol{\Sigma} = \begin{bmatrix} \sigma_{ij} \end{bmatrix} \in \mathbb{R}^{p \times p}$ and $\mathrm{Cov}(\boldsymbol{\epsilon}) = \boldsymbol{\Omega} = \mathrm{Diag}(\omega_{11}, \ldots, \omega_{pp})$. Here $\boldsymbol{\Omega}$ is diagonal since all additive noise variables are assumed to be mutually independent.

For any DAG, if we ignore the directions of all its directed edges, the resulting undirected graph is referred to as the *skeleton* of the DAG. A polytree is a connected DAG whose skeleton does not possess any undirected cycles. The model (2) is referred to as a linear polytree SEM, if the underlying DAG is a polytree $T = (V, E)$. In this paper, we focus on the case of independent Gaussian noise $\epsilon_i$, so the model (2) can be also referred to as a Gaussian linear polytree SEM. A major purpose of this paper is to study the problem of polytree learning, i.e., the recovery of the equivalence class of the polytree $T = (V, E)$ under the model (2) from a finite sample of observations $\boldsymbol{x}_1, \ldots, \boldsymbol{x}_n$, or equivalently the $n \times p$ data matrix $\boldsymbol{X} = [\boldsymbol{x}_1, \ldots, \boldsymbol{x}_n]^\top$. We explain the concept of Markov equivalence classes in the next subsection.

## 2.2 Markov Equivalence and CPDAG

Let us briefly review the concept of Markov equivalence of DAGs. Note that each DAG $G$ entails a list of statements of conditional independence, which are satisfied by any joint distribution Markov to $G$. Two DAGs are equivalent if they entail the same list of conditional independencies. In the present paper, the recovery of the equivalence class of DAG hinges on a well-known result given in Verma & Pearl (1991): Two DAGs are Markov equivalent if and only if they have the same skeleton and sets of v-structures, where a v-structure is a node triplet $i \to k \leftarrow j$ where $i$ and $j$ are non-adjacent.

An important concept to intuitively capture equivalence classes of DAGs is the completed partially DAG (CPDAG): a graph $K$ with both directed and undirected edges representing the Markov equivalence class of a DAG $G$ if: (1) $K$ and $G$ have the same skeleton; (2) $K$ contains a directed edge $i \to j$ if and only if any DAG $G'$ that is Markov equivalent to $G$ contains the same directed edge $i \to j$. The CPDAG of $G$ is denoted as $K = C_G$. It has been shown in Chickering (2002a) that two DAGs have the same CPDAG if and only if they belong to the same Markov equivalence class.

The following result provides some intuition on the CPDAG for polytree models.

**Proposition 1.** *The undirected sub-graph containing undirected edges of the CPDAG of a polytree forms a forest. All equivalent DAGs can be obtained by orienting each undirected tree of the forest into a rooted tree, that is, by selecting any node as the root and setting all edges going away from it.*

*Proof.* Each connected component of the undirected edges is a sub-graph of the polytree $G$'s skeleton, thus is a tree. If a node of the tree also has directed edges, they must be outgoing according to Line 6 of Algorithm 2 (Rule 1 in Meek (1995)). This means that when we convert each undirected tree into a rooted tree, it does not create any additional v-structures in the resulting DAG $G'$. So the original CPDAG is also the CPDAG of $G'$, i.e., $G'$ is equivalent to $G$. On the other hand, if $G'$ is an equivalent DAG, for each undirected tree $T$ in the CPDAG, let $i$ be a source node of $T$ according to $G'$. Then $T$ in $G'$ must be a rooted tree with $i$ being the root to avoid having v-structures within $T$ (and hence contradicting with $G'$ shares the same CPDAG). This shows that all equivalent class members can be obtained by orienting undirected trees into rooted trees and completes the proof. $\square$

## 2.3 Polytree Learning

The procedure of polytree learning we are considering in this paper has been in principle introduced in Rebane & Pearl (1987). The key idea is to first recover the skeleton of the polytree by applying the Chow-Liu algorithm (Chow & Liu, 1968) to the pairwise sample correlations of the data matrix. After the skeleton is recovered, we propose to recover the set of all v-structures via a simple thresholding approach to pairwise sample correlations. Finally, we recover the CPDAG by applying Rule 1 introduced in Verma & Pearl (1992) and justified theoretically in Meek (1995).

### 2.3.1 Chow-Liu Algorithm for Skeleton Recovery

The Chow-Liu tree associated with pairwise correlations, which is the estimated skeleton of the underlying polytree, is defined below.

**Definition 2** (Chow-Liu tree associated to pairwise sample correlations). *Consider the linear polytree model (2) associated to a polytree $T = (V, E)$, whose skeleton is denoted as $\mathcal{T} = (V, \mathcal{E})$. Let $\mathbb{T}_p$ denote the set of undirected trees over $p$ nodes. Given the data matrix $\boldsymbol{X} = [\boldsymbol{x}_1, \ldots, \boldsymbol{x}_n]^\top \in \mathbb{R}^{n \times p}$, we obtain the sample correlation $\hat{\rho}_{ij}$ between $X_i$ and $X_j$ for all $1 \le i < j \le p$. The Chow-Liu tree associated with the pairwise sample correlations is defined as the maximum-weight spanning tree over the $p$ nodes where the weights are absolute values of sample correlations:*

$$\widehat{\mathcal{T}} = \underset{\mathcal{T}=(V,\mathcal{E})\in\mathbb{T}_p}{\arg\max} \sum_{i-j\in\mathcal{E}} |\hat{\rho}_{ij}|. \tag{3}$$

For tree-structured undirected graphical models, it has been established in Chow & Liu (1968) that the maximum likelihood estimation of the underlying tree structure is the Chow-Liu tree associated with the empirical mutual information quantities (which are used to find the maximum-weight spanning tree). The rationale of applying Chow-Liu algorithm to polytree learning has been carefully explained in Rebane & Pearl (1987), to which interested readers are referred. The step of skeleton recovery can be summarized in Algorithm 1.

---

**Algorithm 1** Chow-Liu algorithm

---

    **Input:** The data matrix $\boldsymbol{X} = [\boldsymbol{x}_1, \ldots, \boldsymbol{x}_n]^\top$.
    **Output:** Estimated skeleton $\widehat{\mathcal{T}}$.
  1: Compute the pairwise sample correlations $\hat{\rho}_{ij}$ for all $1 \le i < j \le p$;
  2: Construct a maximum-weight spanning tree using $|\hat{\rho}_{ij}|$ as the edge weights, i.e., $\widehat{\mathcal{T}}$ defined in (3).

---

It is noteworthy that Algorithm 1 can be implemented efficiently by applying Kruskal's algorithm (Kruskal, 1956) to pairwise sample correlations $|\hat{\rho}_{ij}|$ for the construction of maximum weight spanning tree. The computational complexity for Kruskal's algorithm is known to be $O(p^2 \log p)$, which is generally no larger than that for computing the sample correlations, which is $O(p^2 n)$.

### 2.3.2 CPDAG Recovery

In the second part of the procedure of polytree learning, we aim to estimate the CPDAG of the polytree model. Intuitively speaking, this amounts to figuring out all the edges whose orientations can be determined. The first step of this part is to identify all the v-structures. Under the polytree model, any pair of non-adjacent nodes $i$ and $j$ with common neighbor $k$ form a v-structure $i \to k \leftarrow j$ if and only if $X_i$ and $X_j$ are mutually independent. We thus determine the existence of a v-structure $i \to k \leftarrow j$ when the sample correlation $|\hat{\rho}_{ij}| < \rho_{crit} = \gamma_{crit}\sqrt{(\log p)/n}$, where the choice of threshold or critical value is discussed in subsequent sections. After recovering all the v-structures, as aforementioned, it is guaranteed in Meek (1995) that the CPDAG of the polytree can be recovered by iteratively applying the four rules originally introduced in Verma & Pearl (1992). However, given our discussion is restricted to the polytree models, Rules 2, 3, and 4 in Verma & Pearl (1992) and Meek (1995) do not apply. We only need to apply Rule 1 repeatedly. This rule can be stated as follows: Orient any undirected edge $j - k$ into $j \to k$ whenever there is a directed edge $i \to j$ coming from a third node $i$.

These two steps in the second part of polytree structure learning are summarized as Algorithm 2.

## 3 Main Results for Polytree Learning

In this section, we discuss sample size conditions for the recovery of skeleton and CPDAG under a Gaussian linear polytree model $T = (V, E)$. We first establish a correlation decay property on the polytree skeleton (Lemma 3) by applying the famous Wright's formula.

---

**Algorithm 2** Extending the skeleton to a CPDAG

---

    **Input:** Estimated skeleton $\widehat{\mathcal{T}}$, sample correlations $\hat{\rho}_{ij}$'s, tuning parameter $\gamma_{crit}$.

    **Output:** Estimated CPDAG $\widehat{\mathcal{C}}_T$.

1: **for** Each pair of non-adjacent variables $i$, $j$ with common neighbor $k$ in $\widehat{\mathcal{T}}$ **do**
2:     **if** $|\hat{\rho}_{ij}| < \gamma_{crit}\sqrt{(\log p)/n}$ **then**
3:         replace $i - k - j$ with the v-structure $i \to k \leftarrow j$
4:     **end if**
5: **end for**
6: In the resulting graph, orient as many undirected edges as possible by repeatedly applying the rule: orient an undirected edge $j - k$ into $j \to k$ whenever there is a directed edge $i \to j$ for some $i$.

---

### 3.1 Preliminaries

First, the polytree learning method introduced in the previous section depends solely on the marginal correlation coefficients, and is thereby invariant to scaling. Therefore, without loss of generality, we can assume that $X_j$'s have a unit variance for all $j \in V$, i.e. $\boldsymbol{\Sigma}$ is the correlation matrix. It is obvious that the standardized version of a linear SEM is still a linear SEM, and they share the same polytree structure. In this case, by denoting the pairwise correlations as $\rho_{ij} := \text{corr}(X_i, X_j)$, we have $\sigma_{ij} = \rho_{ij}$ for all $1 \leq i, j \leq p$.

Under the linear SEM, we know that $\boldsymbol{B}$ is permutationally similar to a strictly upper triangular matrix, which implies that all eigenvalues of $\boldsymbol{I} - \boldsymbol{B}$ are 1's, and further implies that $\boldsymbol{I} - \boldsymbol{B}$ is invertible. Then, $(\boldsymbol{I} - \boldsymbol{B})^\top \boldsymbol{x} = \boldsymbol{\epsilon}$ implies $\boldsymbol{x} = (\boldsymbol{I} - \boldsymbol{B})^{-\top} \boldsymbol{\epsilon}$, and further implies that $\boldsymbol{x}$ is mean-zero, and has covariance

$$\boldsymbol{\Sigma} = (\boldsymbol{I} - \boldsymbol{B})^{-\top} \boldsymbol{\Omega} (\boldsymbol{I} - \boldsymbol{B})^{-1}.$$

This suggests that we can represent the entries of $\boldsymbol{\Sigma}$ by $(\beta_{ij})$ and $(\omega_{ii})$. In fact, this can be conveniently achieved by using Wright's path tracing formula (Wright, 1960). We first introduce some necessary definitions in order to obtain such expressions. A *trek* connecting nodes $i$ and $j$ in a directed graph $G = (V, E)$ is a sequence of non-colliding consecutive edges connecting $i$ and $j$ of the form

$$i = v_l^L \leftarrow v_{l-1}^L \leftarrow \cdots \leftarrow v_1^L \leftarrow v_0 \to v_1^R \to \cdots \to v_{r-1}^R \to v_r^R = j.$$

We define the left-hand side of $\tau$ as $Left(\tau) = v_l^L \leftarrow \cdots \leftarrow v_0$, the right-hand side of $\tau$ as $Right(\tau) = v_0 \to \cdots \to v_r^R$, and the head of $\tau$ as $H_\tau = v_0$. A trek $\tau$ is said to be a *simple trek* if $Left(\tau)$ and $Right(\tau)$ do not have common edges. In the polytree case, any two nodes $(i, j)$ are connected by a unique path. Also, Wright's famous formula has a simple form:

**Lemma 3.** *Consider the linear polytree model* (2) *with the associated polytree $T = (V, E)$ over $p$ nodes. Also assume that $X_j$ has a unit variance for all $j \in V$. Then, $\rho_{ij} = \beta_{ij}$ for all $i \to j \in E$. Furthermore, for each pair $(i, j)$, the population correlation coefficient satisfies*

$$\rho_{ij} = \begin{cases} \displaystyle\prod_{s \to t \in \tau_{ij}} \rho_{st} & \text{the path connecting $i$ and $j$ is a simple trek} \\ 0 & \text{otherwise.} \end{cases} \tag{4}$$

*Also, the noise variances satisfy*

$$\omega_{jj} = 1 - \sum_{i \in Pa(j)} \rho_{ij}^2, \quad j = 1, \ldots, p. \tag{5}$$

**Remark 4.** *The assumption that the variables $X_1, \ldots, X_p$ have unit variances is unnecessary for* (4) *alone, since correlation coefficients are invariant under standardization.*

**Remark 5.** *Here Eqn.* (5) *can be derived by the following simple argument: Since $T$ is a polytree, all variables in $Pa(j)$ are independent and are also independent with $\epsilon_j$. Evaluating the variance on both sides of $X_j = \sum_{i \in Pa(j)} \beta_{ij} X_i + \epsilon_j$ leads to* (5).

We now introduce the following definitions.

**Definition 6.** *In a standardized linear polytree model (2), let $\rho_{\min}$ and $\rho_{\max}$ be the minimum and maximum absolute correlation over the tree skeleton, that is*

$$\rho_{\min} := \min_{i \to j \in E} |\rho_{ij}|, \quad \rho_{\max} := \max_{i \to j \in E} |\rho_{ij}|.$$

It is noteworthy that in general we cannot assume that $\rho_{\min}$ is independent of $n$ or $p$. In fact, Eqn. (5) gives rise to the following relationship between the noise variance and the correlation coefficients with parents for each node: $\sum_{i \in Pa(j)} \rho_{ij}^2 < 1$, which further implies the following corollary.

**Corollary 7.** *Let $d_*$ represent the highest in-degree for a polytree. Then $\rho_{\min} < \frac{1}{\sqrt{d_*}}$.*

**Remark 8.** *In contrast, it is reasonable to assume $\rho_{\min}$ to be a positive constant independent of $p$ under the undirected tree-structured Gaussian graphical model, since after transforming it to a rooted tree as in Section 2.1, the highest in-degree satisfies $d_* = 1$.*

A key lemma is the following well-known convergence rate for estimating the population correlation matrix:

**Lemma 9.** *Consider a Gaussian linear SEM (2) with $n \geq C_0 \log p$ for some numerical constant $C_0$. Then, on an event $\mathcal{E}$ with probability at least $1 - 1/p^3$, the following inequality holds for some absolute constant $C$:*

$$\|\widehat{\boldsymbol{\rho}} - \boldsymbol{\rho}\|_{\max} < C\sqrt{\frac{\log p}{n}},$$

*where $\boldsymbol{\rho}$ and $\widehat{\boldsymbol{\rho}}$ denote the population and sample correlation matrices, respectively, and $\|\cdot\|_{\max}$ represents the entrywise supremum norm.*

*Proof.* This is a well-known result, which can be obtained by combining Remark 5.40 of Vershynin (2012) and Lemma 1 in Kalisch & Bühlman (2007) (reproduced in Appendix A). See also the proof of the generalized version of this result, Lemma 26). $\qquad\square$

### 3.2 Skeleton Recovery

First, we introduce an important result in analyzing the Chow-Liu algorithm:

**Lemma 10** (e.g. Bresler & Karzand (2020), Lemma 6.1 and Lemma 8.8)**.** *Let $\mathcal{T}$ be the skeleton of true polytree $T = (V, E)$ and $\widehat{\mathcal{T}}$ be the estimated tree through Chow-Liu algorithm (3). If an edge $(w, \tilde{w}) \in \mathcal{T}$ and $(w, \tilde{w}) \notin \widehat{\mathcal{T}}$, i.e. this edge is incorrectly missed, then there exists an edge $(v, \tilde{v}) \in \widehat{\mathcal{T}}$ and $(v, \tilde{v}) \notin \mathcal{T}$ such that $(w, \tilde{w}) \in path_{\mathcal{T}}(v, \tilde{v})$ and $(v, \tilde{v}) \in path_{\widehat{\mathcal{T}}}(w, \tilde{w})$. On such an error event, we have $|\hat{\rho}_{v\tilde{v}}| \geq |\hat{\rho}_{w\tilde{w}}|$.*

We now introduce a sufficient condition on the sample size for skeleton recovery under the Gaussian linear polytree model, in which the independent noise variables satisfy $\epsilon_j \sim \mathcal{N}(0, \omega_{jj})$ for $j = 1, \ldots, p$. Then by $\boldsymbol{x} = (\boldsymbol{I} - \boldsymbol{B})^{-\top}\boldsymbol{\epsilon}$, we know that $\boldsymbol{x}$ is also multivariate Gaussian. This fact will help quantify the discrepancy between population and sample pairwise correlations as characterized in Lemma 9.

**Theorem 11.** *Consider a Gaussian linear SEM (2) associated to a polytree $T = (V, E)$ with $\rho_{\max} < 1 - \delta$. Denote by $\widehat{\mathcal{T}}$ the estimated skeleton by the Chow-Liu algorithm (Algorithm 1), and by $\mathcal{T}$ the true polytree skeleton. Then, on the event $\mathcal{E}$ with probability at least $1 - 1/p^3$ defined in Lemma 9, we have exact polytree skeleton recovery $\widehat{\mathcal{T}} = \mathcal{T}$ as long as*

$$n > \left(\frac{4C^2}{\delta^2}\right)\frac{\log p}{\rho_{\min}^2} \tag{6}$$

*where $C$ is as defined in Lemma 9.*

*Proof.* Consider any undirected edge $(w, \tilde{w}) \in \mathcal{T}$ and any non-adjacent pair $(v, \tilde{v})$ such that $(w, \tilde{w}) \in path_T(v, \tilde{v})$, where $path_T(v, \tilde{v})$ is the path connecting $v$ and $\tilde{v}$ in the polytree $T$. If $path_T(v, \tilde{v})$ is a simple trek, then the correlation decay property, Lemma 3, implies that $\rho_{v\tilde{v}}$ consists of the product among several

correlation coefficients containing $\rho_{w\tilde{w}}$. Hence $|\rho_{v\tilde{v}}| \leq |\rho_{w\tilde{w}}|\rho_{\max}$. On the other hand, if $path_T(v, \tilde{v})$ is not a simple trek, then we have $\rho_{v\tilde{v}} = 0$. Overall, we can obtain an upper bound for $|\rho_{v\tilde{v}}| - |\rho_{w\tilde{w}}|$.

$$|\rho_{v\tilde{v}}| - |\rho_{w\tilde{w}}| \leq |\rho_{w\tilde{w}}|(\rho_{\max} - 1) \leq -\delta\rho_{\min}.$$

Then, on the event $\mathcal{E}$, uniformly for any undirected edge $(w, \tilde{w}) \in \mathcal{T}$ and any non-adjacent pair $(v, \tilde{v})$ such that $(w, \tilde{w}) \in path_{\mathcal{T}}(v, \tilde{v})$, there holds

$$|\hat{\rho}_{v\tilde{v}}| - |\hat{\rho}_{w\tilde{w}}| \leq |\hat{\rho}_{v\tilde{v}} - \rho_{v\tilde{v}}| + |\hat{\rho}_{w\tilde{w}} - \rho_{w\tilde{w}}| + |\rho_{v\tilde{v}}| - |\rho_{w\tilde{w}}|$$
$$< 2C\sqrt{(\log p)/n} - \delta\rho_{\min} < 0,$$

where the last inequality is due to the condition (6). Then, Lemma 10 implies that $\widehat{\mathcal{T}} = \mathcal{T}$ on the event $\mathcal{E}$. $\square$

**Remark 12.** *Our argument on skeleton recovery basically follows the standard arguments based on correlation decay over skeleton in the literature of undirected tree learning, e.g., Nikolakakis et al. (2019); Tavassolipour et al. (2018); Bresler & Karzand (2020). On the other hand, treating undirected tree models as rooted polytree models, our sample size condition for skeleton recovery in Theorem 11 is $n = O(\log p)$, which is the optimal sample size condition for undirected tree recovery.*

**Remark 13.** *The above condition implies some dependence of the sample size on the maximum in-degree $d_*$. In fact, together with Corollary 7, the sample size condition is essentially $n \geq O(d_* \log p)$ if $\rho_{\min} \asymp 1/\sqrt{d_*}$.*

## 3.3 CPDAG Recovery

As described in Section 2.3.2, after obtaining the estimated skeleton, the next step is to identify all v-structures by comparing $\rho_{ij}$ for all node triplets $i - k - j$ in the skeleton with a threshold $\rho_{crit}$. Then the orientation propagation rule described in Algorithm 2 can be applied iteratively to orient as many undirected edges as possible. If both the skeleton and v-structures are correctly identified, the orientation rule will be able to recover the true CPDAG, i.e. the equivalence class (Meek, 1995).

**Theorem 14.** *Consider a Gaussian linear SEM (2) associated to a polytree $T = (V, E)$ with $\rho_{\max} < 1 - \delta$. Denote by $\widehat{\mathcal{T}}$ and $\widehat{\mathcal{C}}_T$ the estimated polytree skeleton from Algorithm 1 and CPDAG from Algorithm 2 with threshold $\gamma_{crit}\sqrt{(\log p)/n}$. Also, denote by $\mathcal{T}$ and $\mathcal{C}_T$ the true polytree skeleton and the true polytree CPDAG, respectively. Then, on the event $\mathcal{E}$ with probability at least $1 - 1/p^3$ defined in Lemma 9, we have exact polytree skeleton recovery $\widehat{\mathcal{T}} = \mathcal{T}$ as well as exact polytree CPDAG recovery $\widehat{\mathcal{C}}_T = \mathcal{C}_T$, as long as*

$$\gamma_{crit} > C \quad and \quad n > C_0(\delta)\gamma_{crit}^2 \left(\frac{\log p}{\rho_{\min}^4}\right), \tag{7}$$

*where $C$ is as defined in Lemma 9, and $C_0(\delta)$ is a constant only depending on $\delta$.*

*Proof.* Since the skeleton recovery is guaranteed by Theorem 11, it suffices to show that under the condition (7), all v-structures are correctly identified on the event $\mathcal{E}$. Let's consider all node triplets $i - k - j$ in $\mathcal{T}$. If the ground truth is $i \rightarrow k \leftarrow j$, we know that $\rho_{ij} = 0$. Then, by (7), on $\mathcal{E}$ we have

$$|\hat{\rho}_{ij}| \leq C\sqrt{(\log p)/n} < \gamma_{crit}\sqrt{(\log p)/n}.$$

This means the v-structure is identified by Algorithm 2.

In contrast, if the ground truth is $i \leftarrow k \leftarrow j$ or $i \leftarrow k \rightarrow j$ or $i \rightarrow k \rightarrow j$, the correlation decay property Lemma 3 implies that $|\rho_{ij}| = |\rho_{ik}||\rho_{kj}| \geq \rho_{\min}^2$. Then, on $\mathcal{E}$, there holds

$$|\hat{\rho}_{ij}| \geq |\rho_{ij}| - C\sqrt{(\log p)/n} \geq \rho_{\min}^2 - C\sqrt{(\log p)/n} > \gamma_{crit}\sqrt{(\log p)/n},$$

where the last inequality is also due to (7). This means this triplet is correctly identified as a non-v-structure.

In sum, we identify all the v-structures exactly. Then the CPDAG of $T$ can be exactly recovered by Algorithm 2 as guaranteed in Meek (1995). $\square$

**Remark 15.** *It is noteworthy to observe the difference between the sample size conditions in Theorems 11 and 14. In particular, if $\rho_{\min} \asymp 1/\sqrt{d_*}$, the above sufficient condition on sample size for CPDAG recovery is essentially $n \geq O(d_*^2 \log p)$, while recall that the sample size condition for skeleton recovery is $n \geq O(d_* \log p)$. This dependence on maximum in-degree is probably a particular property for polytree learning, given that most existing theory on general sparse DAG recovery usually requires the sample size to be greater than the maximum neighborhood size, e.g., Theorem 2 in Kalisch & Bühlman (2007).*

### 3.4 Information-theoretic Lower Bounds on the Sample Size

In this subsection, we will establish the necessary conditions on the sample size for both skeleton and CPDAG recovery under Gaussian linear polytree models. In particular, we will use Fano's method to derive information-theoretic bounds.

**Theorem 16.** *Let $\mathbb{T}(\rho_{\min})$ be a collection of Gaussian linear polytree models, such that $\rho_{\min} := \min_{i \to j \in E} |\rho_{ij}|$ is fixed and satisfies $0 < \rho_{\min} < 1/\sqrt{p}$. In each model out of this class, assume that $\rho_{\max} := \max_{i \to j \in E} |\rho_{ij}| < 1/2$. Assume $p \geq 10$. Suppose that $\mathbb{T}(\rho_{\min})$ is indexed by $\theta$, with corresponding polytree $T_\theta$, covariance matrix $\Sigma_\theta$, tree skeleton $\mathcal{T}_\theta$, and CPDAG $\mathcal{C}_{T_\theta}$. Then for any skeleton estimator $\widehat{\mathcal{T}}$, there holds*

$$\sup_{\theta \in \mathbb{T}(\rho_{\min})} \mathbb{P}_{\Sigma_\theta}(\widehat{\mathcal{T}}(X) \neq \mathcal{T}_\theta) \geq 1/2$$

*provided*

$$n < \frac{1}{\rho_{\min}^2}(\log(p-2) - 2).$$

*Moreover, for any CPDAG estimator $\widehat{\mathcal{C}}$, there holds*

$$\sup_{\theta \in \mathbb{T}(\rho_{\min})} \mathbb{P}_{\Sigma_\theta}(\widehat{\mathcal{C}}(X) \neq \mathcal{C}_{T_\theta}) \geq 1/2$$

*provided*

$$n < \frac{1}{5\rho_{\min}^4}\left(\log \frac{(p-1)(p-2)}{2} - 2\right).$$

*Proof.* The key idea is to apply Fano's method to appropriate sub-classes of $\mathbb{T}(\rho_{\min})$ to establish the intended information-theoretic lower bounds for both skeleton and CPDAG recovery. Generally speaking, let $\mathbb{T}_M = \{T_1, \ldots, T_M\}$ be a sub-class of polytree models $\mathbb{T}(\rho_{\min})$ whose respective covariance matrices are denoted as $\Sigma(T_1), \ldots, \Sigma(T_M)$. Let model index $\theta$ be chosen uniformly at random from $\{1, \ldots, M\}$. Given the observations $X \in \mathbb{R}^{n \times p}$, the decoder $\psi$ estimates the underlying polytree structure with maximal probability of decoding error defined as

$$p_{\text{err}}(\psi) = \max_{1 \leq j \leq M} \mathbb{P}_{\Sigma(T_j)}(\psi(X) \neq T_j).$$

Fano's inequality (Thomas & Joy, 2006) shows that the maximal probability of error over $\mathbb{T}_M$, $p_{\text{err}}(\psi)$, can be lower bounded as

$$\inf_\psi p_{\text{err}}(\psi) \geq 1 - \frac{I(\theta; X) + 1}{\log M}.$$

Given all involved distributions are multivariate Gaussian, we will apply the following entropy-based bound of the mutual information that can be found in Wang et al. (2010):

$$I(\theta; X) \leq \frac{n}{2}F(\mathbb{T}), \quad \text{where}$$

$$F(\mathbb{T}) := \log\det(\overline{\Sigma}) - \frac{1}{M}\sum_{j=1}^M \log\det(\Sigma(T_j)) \tag{8}$$

and the averaged covariance matrix $\overline{\Sigma} := \frac{1}{M}\sum_{j=1}^M \Sigma(T_j)$.

**Lower Bound for Skeleton Recovery**

In the following we consider a class of polytree models $\mathbb{T}_M = \{T_1, \ldots, T_M\}$ where $M = p-2$. These polytrees share $p-2$ common directed edges $1 \to (p-1)$, $2 \to (p-1)$, ..., $(p-2) \to (p-1)$. For the $(p-1)$-th directed edge, we let $p \to 1$ in $T_1$, $p \to 2$ in $T_2$, ..., $p \to (p-2)$ in $T_{p-2}$. Also, we assume that all variables have variance one, and the correlation coefficients on the skeleton are all $\rho$ that satisfies $0 < \rho < \frac{1}{\sqrt{p}}$. Here we write $\rho = \rho_{\min}$ for simplicity. Note that the polytrees in this sub-class of $\mathbb{T}(\rho)$ (defined in the statement of Theorem 16) have distinct skeletons, so

$$\inf_{\widehat{\mathcal{T}}} \sup_{\theta \in \mathbb{T}(\rho_{\min})} \mathbb{P}_{\boldsymbol{\Sigma}_\theta}(\widehat{\mathcal{T}}(\boldsymbol{X}) \neq \mathcal{T}_\theta) \geq \inf_{\psi} \max_{1 \leq j \leq M} \mathbb{P}_{\boldsymbol{\Sigma}(T_j)}\left(\psi(\boldsymbol{X}) \neq T_j\right).$$

We can easily obtain the formula for each covariance $\boldsymbol{\Sigma}(T_j)$ for $j = 1, \ldots, M$ by using Lemma 3. For example, for $T_1$, we have

$$\boldsymbol{\Sigma}(T_1) = \begin{bmatrix} 1 & 0 & \ldots & 0 & \rho & \rho \\ 0 & 1 & \ldots & 0 & \rho & 0 \\ \vdots & \vdots & \ddots & \vdots & \vdots & \vdots \\ 0 & 0 & \ldots & 1 & \rho & 0 \\ \hdashline \rho & \rho & \ldots & \rho & 1 & \rho^2 \\ \rho & 0 & \ldots & 0 & \rho^2 & 1 \end{bmatrix} := \begin{bmatrix} \boldsymbol{A} & \boldsymbol{B} \\ \boldsymbol{B}^\top & \boldsymbol{D} \end{bmatrix}$$

The Schur complement of $\boldsymbol{A} = \boldsymbol{I}$ is thereby

$$\boldsymbol{D} - \boldsymbol{B}^\top \boldsymbol{A}^{-1} \boldsymbol{B} = \begin{bmatrix} 1 - (p-2)\rho^2 & 0 \\ 0 & 1 - \rho^2 \end{bmatrix}.$$

Then

$$\det(\boldsymbol{\Sigma}(T_1)) = \det(\boldsymbol{A}) \det(\boldsymbol{D} - \boldsymbol{B}^\top \boldsymbol{A}^{-1} \boldsymbol{B}) = (1 - \rho^2)(1 - (p-2)\rho^2).$$

Similarly, for all $j = 1, \ldots, p-2$, there holds $\det(\boldsymbol{\Sigma}(T_j)) = (1 - \rho^2)(1 - (p-2)\rho^2)$.

On the other hand, the average covariance is

$$\overline{\boldsymbol{\Sigma}} = \frac{1}{p-2} \sum_{j=1}^{p-2} \boldsymbol{\Sigma}(T_j) = \begin{bmatrix} 1 & 0 & \ldots & 0 & \rho & \rho/(p-2) \\ 0 & 1 & \ldots & 0 & \rho & \rho/(p-2) \\ \vdots & \vdots & \ddots & \vdots & \vdots & \vdots \\ 0 & 0 & \ldots & 1 & \rho & \rho/(p-2) \\ \hdashline \rho & \rho & \ldots & \rho & 1 & \rho^2 \\ \rho/(p-2) & \rho/(p-2) & \ldots & \rho/(p-2) & \rho^2 & 1 \end{bmatrix}.$$

As with above, we can use Schur complement to obtain

$$\det(\overline{\boldsymbol{\Sigma}}) = (1 - \rho^2/(p-2))(1 - (p-2)\rho^2).$$

Plug these results into (8), we have

$$F(\mathbb{T}) = \log\left(1 + \frac{(p-3)\rho^2}{(p-2)(1-\rho^2)}\right) \leq \frac{(p-3)\rho^2}{(p-2)(1-\rho^2)} \leq \frac{(p-3)\rho^2}{(p-2)(1-1/p)} \leq \rho^2.$$

Then $p_{\text{err}} \geq 1 - (\frac{n}{2}\rho^2 + 1)/\log(p-2)$. To ensure $p_{\text{err}} > 1/2$, we only need to require $1 - (\frac{n}{2}\rho^2 + 1)/\log(p-2) > 1/2$, which is equivalent to $n < (\log(p-2) - 2)/\rho^2$.

**Lower Bound for CPDAG Recovery**

Let's now consider another class of polytree models $\mathbb{T}_M = \{T_1, \ldots, T_M\}$ where $M = \binom{p-1}{2}$. All polytrees in this class are stars with hub node $p$, and $p$ is directed to all but two nodes in $\{1, \ldots, p-1\}$. In $T_1$, the directed edges are $1 \to p$, $2 \to p$, $p \to 3$, $p \to 4$, ..., $p \to (p-1)$. In $T_2$, the directed edges are $1 \to p$, $p \to 2$,

$3 \to p$, $p \to 4$, ..., $p \to (p-1)$. And so on until in $T_M$, the directed edges are $p \to 1$, $p \to 2$, ..., $p \to (p-3)$, $(p-2) \to p$, $(p-1) \to p$. Also, assume that all variables have variance one, and the correlation coefficients on the skeleton are all $\rho$ that satisfies $0 < \rho < \frac{1}{2}$. Again, we write $\rho = \rho_{\min}$ for simplicity. Although the polytrees in this sub-class of $\mathbb{T}(\rho)$ have the same skeletons, but they have distinct CPDAGs since they have distinct sets of v-structures. Therefore,

$$\inf_{\widehat{C}} \sup_{\theta \in \mathbb{T}(\rho_{\min})} \mathbb{P}_{\boldsymbol{\Sigma}_\theta}(\widehat{\mathcal{C}}(\boldsymbol{X}) \neq \mathcal{C}_{T_\theta}) \geq \inf_{\psi} \max_{1 \leq j \leq M} \mathbb{P}_{\boldsymbol{\Sigma}(T_j)}(\psi(\boldsymbol{X}) \neq T_j).$$

Again, we have the formula for each covariance $\boldsymbol{\Sigma}(T_j)$ for $j = 1, \ldots, M$ by using Lemma 3. For example, for $T_1$, we have

$$\boldsymbol{\Sigma}(T_1) = \begin{bmatrix} 1 & 0 & \rho^2 & \cdots & \rho^2 & \rho \\ 0 & 1 & \rho^2 & \cdots & \rho^2 & \rho \\ \rho^2 & \rho^2 & 1 & \cdots & \rho^2 & \rho \\ \vdots & \vdots & \vdots & \ddots & \vdots & \vdots \\ \rho^2 & \rho^2 & \rho^2 & \cdots & 1 & \rho \\ \rho & \rho & \rho & \cdots & \rho & 1 \end{bmatrix}$$

Recall that in a linear polytree model there holds $\boldsymbol{\Sigma} = (\boldsymbol{I} - \boldsymbol{B})^{-\top}\boldsymbol{\Omega}(\boldsymbol{I} - \boldsymbol{B})$. Since $\boldsymbol{B}$ can be transformed to a strict upper triangular matrix by permuting the $p$ nodes, we know that $\det(\boldsymbol{I} - \boldsymbol{B}) = 1$. Then

$$\det(\boldsymbol{\Sigma}) = \det(\boldsymbol{\Omega}) = \prod_{j=1}^p \omega_{jj} = \prod_{j=1}^p \left(1 - \sum_{i \in Pa(j)} \rho_{ij}^2\right).$$

Then for $j = 1, \ldots, M$, there holds $\det(\boldsymbol{\Sigma}(T_j)) = (1 - \rho^2)^{p-3}(1 - 2\rho^2)$, which implies that

$$\text{logdet}(\boldsymbol{\Sigma}(T_j)) = (p-3)\log(1 - \rho^2) + \log(1 - 2\rho^2).$$

On the other hand, we have

$$\overline{\boldsymbol{\Sigma}} = \frac{1}{M}\sum_{j=1}^M \boldsymbol{\Sigma}(T_j) = \begin{bmatrix} 1 & \frac{M-1}{M}\rho^2 & \cdots & \frac{M-1}{M}\rho^2 & \rho \\ \frac{M-1}{M}\rho^2 & 1 & \cdots & \frac{M-1}{M}\rho^2 & \rho \\ \vdots & \vdots & \ddots & \vdots & \vdots \\ \frac{M-1}{M}\rho^2 & \frac{M-1}{M}\rho^2 & \cdots & 1 & \rho \\ \rho & \rho & \cdots & \rho & 1 \end{bmatrix} := \begin{bmatrix} \boldsymbol{A} & \boldsymbol{B} \\ \boldsymbol{B}^\top & \boldsymbol{D} \end{bmatrix}.$$

The Schur complement of $\boldsymbol{D} = 1$ is

$$\overline{\boldsymbol{\Sigma}}/\boldsymbol{D} = \boldsymbol{A} - \boldsymbol{B}\boldsymbol{D}^{-1}\boldsymbol{B}^\top == \begin{bmatrix} 1 - \rho^2 & -\rho^2/M & \cdots & -\rho^2/M \\ -\rho^2/M & 1 - \rho^2 & \cdots & -\rho^2/M \\ \vdots & \vdots & \ddots & \vdots \\ -\rho^2/M & -\rho^2/M & \cdots & 1 - \rho^2 \end{bmatrix}.$$

It's easy to obtain all the eigenvalues of $\overline{\boldsymbol{\Sigma}}/\boldsymbol{D}$: $1 - \frac{p+1}{p-1}\rho^2$ with multiplicity 1 and $1 - \frac{p(p-3)}{(p-1)(p-2)}\rho^2$ with multiplicity $p-2$. Plug these results into (8), we have

$$
\begin{aligned}
F(\mathbb{T}) &= \log\left(1 - \frac{p+1}{p-1}\rho^2\right) + (p-2)\log\left(1 - \frac{p(p-3)}{(p-1)(p-2)}\rho^2\right) \\
&\quad - \log(1-2\rho^2) - (p-3)\log(1-\rho^2) \\
&= \log\left(1 - \frac{p+1}{p-1}\rho^2\right) + (p-2)\log\left(1 + \frac{2}{(p-1)(p-2)}\frac{\rho^2}{1-\rho^2}\right) + \log\left(1 + \frac{\rho^2}{1-2\rho^2}\right) \\
&\leq -\frac{p+1}{p-1}\rho^2 + \frac{2}{p-1}\frac{\rho^2}{1-\rho^2} + \frac{\rho^2}{1-2\rho^2} \\
&= -\frac{p+1}{p-1}\rho^2 + \frac{2}{p-1}\left(\rho^2 + \frac{\rho^4}{1-\rho^2}\right) + \rho^2 + \frac{2\rho^4}{1-2\rho^2} \\
&= \frac{2}{p-1}\frac{\rho^4}{1-\rho^2} + \frac{2\rho^4}{1-2\rho^2} < 5\rho^4,
\end{aligned}
$$

where the first inequality is due to $\log(1+x) \leq x$, and the second inequality is due to the assumption that $\rho^2 < 1/4$ and $p \geq 10$. As with the case of skeleton recovery, we know that $p_{\mathrm{err}} > 1/2$ as long as we require that

$$
n < \frac{1}{5\rho^4}\left(\log\frac{(p-1)(p-2)}{2} - 2\right).
$$

$\square$

Compare Theorem 16 with Theorems 11 and 14, we can conclude that our derived sufficient conditions on the sample sizes for the recovery of both skeleton and CPDAG are sharp.

## 4 PC Algorithm Adapted to Polytree Models

In this section, we introduce another algorithm to recover the skeleton and CPDAG of the polytree, which is adapted from the PC algorithm but amenable to linear polytree structure.

To implement a PC algorithm that is adapted to polytree structures, we consider an early stopping version of the algorithm explained in Kalisch & Bühlman (2007). The following lemma demonstrates important properties of marginal and conditional probabilities on a polytree.

**Lemma 17.** *Consider a Gaussian linear SEM* (2) *associated to a polytree* $T = (V, E)$ *with* $\rho_{\max} < 1 - \delta$. *Then we have the following:*

1. *For any* $(i,j) \in \mathcal{T}$, $|\rho_{ij}| \geq \rho_{\min}$ *and* $|\rho_{ij|k}| \geq \delta\rho_{\min}$ *for any* $k \notin \{i,j\}$.

2. *For any* $(i,j) \notin \mathcal{T}$, *if the path connecting* $i$ *and* $j$ *is not a trek, then* $\rho_{ij} = 0$.

3. *For any* $(i,j) \notin \mathcal{T}$, *if the path connecting* $i$ *and* $j$ *is a trek, then there exists some* $k \in \mathrm{adj}(i,\mathcal{T}) \cup \mathrm{adj}(j,\mathcal{T})\backslash\{i,j\}$ *such that* $\rho_{ij|k} = 0$.

*Proof.* We discuss these three cases separately:

1. In this case, we have $|\rho_{ij}| \geq \rho_{\min}$ by definition. For any $k \notin \{i,j\}$, since $(i,j) \in \mathcal{T}$, by the tree structure, we know either $i$ lies in the path connecting $j$ and $k$, or $j$ lies in the path connecting $i$ and $k$. WLOG, we can assume the former is true. Then by the correlation decay property Lemma 3, there holds $|\rho_{jk}| \leq \rho_{\max}|\rho_{ij}|$, which implies $|\rho_{ij} - \rho_{ik}\rho_{jk}| \geq |\rho_{ij}|(1 - \rho_{\max}^2)$. Then

$$
|\rho_{ij|k}| = \left|\frac{\rho_{ij} - \rho_{ik}\rho_{jk}}{\sqrt{(1-\rho_{ik}^2)(1-\rho_{jk}^2)}}\right| \geq \rho_{\min}(1 - \rho_{\max}^2) \geq \delta\rho_{\min}.
$$

2. This is directly implied by Lemma 3.

3. When the path connecting $i$ and $j$ is a trek, there exists some $k \in \text{adj}(i, \mathcal{T}) \cup \text{adj}(j, \mathcal{T}) \backslash \{i, j\}$ on this path. Lemma 3 implies $\rho_{ij} = \rho_{ik}\rho_{jk}$, which implies $\rho_{ij|k} = 0$.

$\square$

Besides the population correlations $\rho_{ij}$, for any distinct $i$, $j$, and $k$, the population and sample partial correlations can be represented by marginal correlations through the following equations:

$$\rho_{ij|k} = \frac{\rho_{ij} - \rho_{ik}\rho_{jk}}{\sqrt{(1 - \rho_{ik}^2)(1 - \rho_{jk}^2)}} \quad \text{and} \quad \hat{\rho}_{ij|k} = \frac{\hat{\rho}_{ij} - \hat{\rho}_{ik}\hat{\rho}_{jk}}{\sqrt{(1 - \hat{\rho}_{ik}^2)(1 - \hat{\rho}_{jk}^2)}}. \tag{9}$$

Note that although the concept of partial correlations will not be used in this section, we need it in later sections. The relationship between population and sample marginal and conditional correlations can be characterized by the following lemma, which is a simplified version of Corollary 1 in Kalisch & Bühlman (2007).

**Lemma 18.** *Consider a Gaussian linear SEM* (2) *with $n \geq C_0 \log p$ for some numerical constant $C_0$. Then, on an event $\mathcal{E}$ with probability at least $1 - 1/p^3$, the following inequality holds for some absolute constant $C$:*

$$\|\widehat{\boldsymbol{\rho}} - \boldsymbol{\rho}\|_{\max} < C\sqrt{\frac{\log p}{n}},$$

*and*

$$\left|\hat{\rho}_{ij|k} - \rho_{ij|k}\right| < C\sqrt{\frac{\log p}{n}} \quad \forall i < j, \ k \notin \{i, j\}.$$

*Proof.* This is an easy generalization of Lemma 9 by the relationship between sample partial correlation and sample correlation cumulative distribution functions under multivariate normal distributions established in Fisher (1924). $\square$

From the above lemma, it is natural to consider the early-stopping PC algorithm with a tuning parameter $\gamma_{crit}$ that is described in Algorithm 3. The estimated skeleton is denoted as $\widehat{\mathcal{T}}$.

---

**Algorithm 3** Estimating the polytree skeleton by the simplified PC algorithm

    **Input:** The $n \times p$ data matrix $\boldsymbol{X}$; tuning parameter $\gamma_{crit}$
    **Output:** Estimated skeleton $\widehat{\mathcal{T}}$.
1: Compute the sample correlations $\hat{\rho}_{ij}$ for all $1 \leq i < j \leq p$;
2: Compute the sample partial correlations $\hat{\rho}_{ij|k}$; for all $1 \leq i < j \leq p$ and any $k \notin \{i, j\}$;
3: The complete undirected graph over the $p$ nodes is denoted as $\mathcal{G}_0$;
4: **for** Each pair of non-ordered $(i, j)$ **do**
5:     **if** $|\hat{\rho}_{ij}| < \gamma_{crit}\sqrt{(\log p)/n}$ **then**
6:         remove $(i, j)$ from $\mathcal{G}_0$
7:     **else if** $|\hat{\rho}_{ij|k}| < \gamma_{crit}\sqrt{(\log p)/n}$ for some $k \notin \{i, j\}$ **then**
8:         remove $(i, j)$ from $\mathcal{G}_0$
9:     **end if**
10: **end for**
11: The resulting graph is denoted as $\widehat{\mathcal{T}}$.

---

Our consistency result for polytree skeleton recovery by the simplified PC algorithm demonstrated in Algorithm 3 consists of two parts. In the first part, we show that $\widehat{\mathcal{T}} \subset \mathcal{T}$ as long as the tuning parameter $\gamma_{crit}$ in the threshold is chosen large enough; in the second part, we show that $\widehat{\mathcal{T}} = \mathcal{T}$ if we assume further that $\rho_{\min}$ satisfies some lower bound condition.

**Theorem 19.** *Consider a Gaussian linear SEM* (2) *associated to a polytree $T = (V, E)$ with $\rho_{\max} < 1 - \delta$. Let $\widehat{\mathcal{T}}$ be the estimated skeleton from the simplified PC algorithm given in Algorithm 3 with the threshold $\gamma_{crit}\sqrt{(\log p)/n}$. If the tuning parameter satisfies $\gamma_{crit} > C$, where $C$ is as defined in Lemma 18, then, on the event $\mathcal{E}$ defined in Lemma 18 with probability at least $1 - 1/p^3$, we have $\widehat{\mathcal{T}} \subset \mathcal{T}$.*

*In addition, if*

$$n > \left( \frac{4\gamma_{crit}^2}{\delta^2} \right) \frac{\log p}{\rho_{\min}^2}, \tag{10}$$

*we have $\widehat{\mathcal{T}} = \mathcal{T}$ on $\mathcal{E}$, i.e. the exact recovery of the polytree skeleton.*

*Proof.* Let the event $\mathcal{E}$ be defined as in Lemma 18. Consider any $(i,j) \notin \mathcal{T}$. If the path connecting $i$ and $j$ is not a trek, we have $\rho_{ij} = 0$ by Lemma 17. Then Lemma 18 implies $|\hat{\rho}_{ij}| < C\sqrt{(\log p)/n} < \gamma_{crit}\sqrt{(\log p)/n}$, so $(i,j)$ is excluded from $\mathcal{G}_0$ by Algorithm 3. On the other hand, if the path connecting $i$ and $j$ is a trek, Lemma 17 implies that there exists some $k \notin \{i,j\}$ such that $\rho_{ij|k} = 0$. Then Lemma 18 implies $|\hat{\rho}_{ij|k}| < C\sqrt{(\log p)/n} < \gamma_{crit}\sqrt{(\log p)/n}$, so $(i,j)$ is also excluded from $\mathcal{G}_0$. This implies any $(i,j) \notin \mathcal{T}$ is removed from $\mathcal{G}_0$, no matter whether the path connecting them is a trek or not. Thus, we have $\widehat{\mathcal{T}} \subset \mathcal{T}$.

Further, the condition (10) implies $\rho_{\min} > \frac{2\gamma_{crit}}{\delta}\sqrt{\frac{\log p}{n}}$, for any $(i,j) \in \mathcal{T}$, Lemmas 18 and 17 imply

$$|\hat{\rho}_{ij}| \geq |\rho_{ij}| - C\sqrt{\frac{\log p}{n}} \geq \rho_{\min} - C\sqrt{\frac{\log p}{n}} > \gamma_{crit}\sqrt{\frac{\log p}{n}},$$

and for any $k \notin \{i,j\}$,

$$|\hat{\rho}_{ij|k}| \geq |\rho_{ij|k}| - C\sqrt{\frac{\log p}{n}} \geq \delta\rho_{\min} - C\sqrt{\frac{\log p}{n}} > \gamma_{crit}\sqrt{\frac{\log p}{n}}.$$

This implies that $(i,j) \in \widehat{\mathcal{T}}$, so $\widehat{\mathcal{T}} = \mathcal{T}$. □

**Remark 20.** *Here we show that the skeleton $\mathcal{T}$ can be exactly recovered with high probability under the sample size condition* (10), *which is comparable to the condition* (6) *for skeleton recovery by the Chow-Liu algorithm.*

Note that our simplified PC algorithm is only aimed at recovering the skeleton rather than identifying all marginal/conditional independence relationships among the variables. This is the reason why the sample size condition (10) can be smaller than the necessary condition for CPDAG recovery established in Theorem 16. This is the crucial difference between Theorem 19 and standard CPDAG recovery result by PC algorithm for sparse DAG learning, e.g. Kalisch & Bühlman (2007).

To understand why Algorithm 3 may lead to consistent skeleton recovery even without identifying the marginal/conditional independence relationships correctly, one can take a trek $i \to k \to j$ in the true polytree with $|\rho_{ik}| = |\rho_{jk}| = \rho_{\min}$ as an example. In this case, Algorithm 3 could possibly remove the edge $i - j$ simply due to $|\hat{\rho}_{ij}| < \gamma_{crit}\sqrt{(\log p)/n}$ as long as $|\rho_{ij}| = |\rho_{ik}||\rho_{jk}| = \rho_{\min}^2$ is sufficiently small. Following the idea of the PC algorithm, one may record $(X_i, X_j)$ as an independent pair of variables incorrectly. However, this may still lead to correct skeleton recovery.

To recover the CPDAG, we can naturally apply Algorithm 2 following Algorithm 3. The following result is obvious and we omit the proof.

**Theorem 21.** *Under the assumptions in Theorem 19, if we further assume the sample size condition* (7) *holds, then Algorithms 3 and 2 recover the true CPDAG exactly with probability at least $1 - 1/p^3$.*

## 5  Inverse Correlation Matrix Estimation

In this section, we are interested in recovering the inverse correlation matrix of the polytree model under a recovered CPDAG. This is particularly useful for likelihood calculation; see, e.g. van de Geer &

Bühlmann (2013). This could be useful for choosing the value of tuning parameters with likelihood-based cross-validation.

To estimate the inverse correlation matrix, due to the scaling invariance of population and sample correlations, without loss of generality, we assume that all $X_i$'s have unit variances. Then the inverse correlation matrix is $\boldsymbol{\Theta} := \boldsymbol{\Sigma}^{-1} = (\boldsymbol{I} - \boldsymbol{B})\boldsymbol{\Omega}^{-1}(\boldsymbol{I} - \boldsymbol{B}^{\top})$. The major goal of this subsection is to study how well we can estimate $\boldsymbol{\Theta}$. It may be noteworthy that the error bound we obtained (Theorem 24) depends on the total number of v-structures in addition to the usual dimension and sample size.

## 5.1 Inverse Correlation Matrix and CPDAG

At first, let's choose one realization from the equivalence class represented by this CPDAG, and still refer to it as $T$ with no confusion. By $\boldsymbol{\Theta} = (\boldsymbol{I} - \boldsymbol{B})\boldsymbol{\Omega}^{-1}(\boldsymbol{I} - \boldsymbol{B}^{\top})$, and the fact $\beta_{ij} = \rho_{ij}$ for each $i \to j \in T$, $\beta_{ij}$ due to unit variances, we can represent the entries of the inverse correlation matrix by the correlation coefficients over the polytree as

$$
\theta_{ij} = \begin{cases} -\rho_{ij}/\omega_{jj} & \text{if } i \to j \in T \\ -\rho_{ji}/\omega_{ii} & \text{if } j \to i \in T \\ \rho_{ik}\rho_{jk}/\omega_{kk} & \text{if } i \to k \leftarrow j \in T \\ 0 & \text{otherwise,} \end{cases} \quad \text{for } i \neq j \tag{11}
$$

$$
\theta_{jj} = \frac{1}{\omega_{jj}} + \sum_{k \in Ch(j)} \frac{\rho_{jk}^2}{\omega_{kk}}, \text{ for } j = 1, \ldots p. \tag{12}
$$

where $\omega_{jj} = 1 - \sum_{i \in Pa(j)} \rho_{ij}^2$ for $j = 1, \ldots, p$. Notice that the $k$ in $i \to k \leftarrow j \in T$ must be unique in a polytree.

A natural question is whether we can represent the inverse correlation matrix only through the CPDAG $\mathcal{C}_T$. This question is important given we can only hope to recover $\mathcal{C}_T$ by the algorithms introduced in Sections 2.3.1 and 2.3.2. We first give a useful lemma, which explains for what kind of node $j$, the noise variance $\omega_{jj} = 1 - \sum_{i \in Pa(j)} \rho_{ij}^2$ is well-defined on the CPDAG $\mathcal{C}_T$, i.e., invariant to any particular polytree chosen from the equivalence class.

**Lemma 22.** *Denote by $\mathcal{C}_T$ the true CPDAG of the polytree $T$. We denote by $V_m$ the collection of nodes $j$ such that there is at least one undirected edge $i - j$ in $\mathcal{C}_T$. On the other hand, we denote $V_d$ the collection of nodes $j$ such that all its neighbors are connected to it with a directed edge in $\mathcal{C}_T$. This means that $V_m$ and $V_d$ form a partition of all nodes. Then, we have the following properties:*

1. *For each $j \in V_m$, there is no $i$ satisfying $i \to j \in \mathcal{C}_T$.*

2. *For each $j \in V_m$ and any polytree $T'$ within the equivalence class $\mathcal{C}_T$, $j$ has at most one parent in $T'$.*

3. *For each $j \in V_d$, since the set of parents of $j$ is determined by the CPDAG $\mathcal{C}_T$, the corresponding noise variance $\omega_{jj} = 1 - \sum_{i \in Pa(j)} \rho_{ij}^2$ is well-defined.*

4. *Combining the third property and the contrapositive of the first property, we know for each $i \to j \in \mathcal{C}_T$, we have $j \in V_d$, and the corresponding noise variance $\omega_{jj}$ is thereby well-defined.*

We omit the proof since this result can be directly implied by the fact that v-structures are kept unchanged in all polytrees within the equivalence class determined by $\mathcal{C}_T$. Then the following result shows that the inverse correlation matrix can be represented by the pairwise correlations on the skeleton as well as the CPDAG.

**Lemma 23.** *Let $V_m$ and $V_d$ be the partition of all nodes defined in Lemma 22. Then, the inverse correlation matrix can be represented as*

$$\theta_{ij} = \begin{cases} -\rho_{ij}/\omega_{jj} & \text{if } i \to j \in \mathcal{C}_T \\ -\rho_{ji}/\omega_{ii} & \text{if } j \to i \in \mathcal{C}_T \\ -\rho_{ij}/(1-\rho_{ij}^2) & \text{if } i - j \in \mathcal{C}_T \\ \rho_{ik}\rho_{jk}/\omega_{kk} & \text{if } i \to k \leftarrow j \in \mathcal{C}_T \\ 0 & \text{otherwise,} \end{cases} \quad \text{for } i \neq j$$

*and*

$$\theta_{jj} = \begin{cases} \frac{1}{\omega_{jj}} + \sum\limits_{j \to k \in \mathcal{C}_T} \frac{\rho_{jk}^2}{\omega_{kk}}, & j \in V_d, \\ 1 + \sum\limits_{j-k \in \mathcal{C}_T} \frac{\rho_{jk}^2}{1-\rho_{jk}^2} + \sum\limits_{j \to k \in \mathcal{C}_T} \frac{\rho_{jk}^2}{\omega_{kk}}, & j \in V_m. \end{cases}$$

*Here $\omega_{jj} = 1 - \sum\limits_{i \in Pa(j)} \rho_{ij}^2$ is well-defined in all of the above formulas, since $Pa(j)$ is well-defined for any $j \in V_d$.*

*Proof.* Let $T'$ be a polytree in the equivalent class of $\mathcal{C}_T$. For $\theta_{ij}$, the possible cases listed in the lemma are exhaustive by Lemma 23. the only case need checking is $i - j \in \mathcal{C}_T$. If the edge is oriented as $i \to j$ in $T'$. By point 2. in Lemma 23, $i$ is the only parent of $j$ thus $w_{jj} = 1 - \rho_{ij}^2$. By Eq. (12), $\theta_{ij} = -\rho_{ij}/(1-\rho_{ij}^2)$. It is easy to see that the result is identical if the undirected edge is $j \to j$ in some polytree.

For $\theta_{jj}$ and the non-obvious case of $j \in V_m$, the undirected edges $j - k$ connected to $j$ in $\mathcal{C}_T$ are either all, or except for one, oriented as $j \to k$ in any polytree $T'$ (Lemma 23). For the first case, $\omega_{jj} = 1$, Eq. (12) in $T'$ becomes

$$\theta_{jj} = 1 + \sum_{j-k \in \mathcal{C}_T} \frac{\rho_{jk}^2}{\omega_{kk}} + \sum_{j \to k \in \mathcal{C}_T} \frac{\rho_{jk}^2}{\omega_{kk}} = 1 + \sum_{j-k \in \mathcal{C}_T} \frac{\rho_{jk}^2}{1-\rho_{jk}^2} + \sum_{j \to k \in \mathcal{C}_T} \frac{\rho_{jk}^2}{\omega_{kk}}.$$

Note that $\omega_{kk} = 1 - \rho_{jk}^2$ according to the result derived above for undirected edges.

In the second case, suppose $k_1$ is the single parent of $j$ in $T'$. We have $\omega_{jj} = 1 - \rho_{jk_1}^2$, and Eq. (12) is

$$\theta_{jj} = \frac{\rho_{jk_1}^2}{1-\rho_{jk_1}^2} + \sum_{k \neq k_1, j-k \in \mathcal{C}_T} \frac{\rho_{jk}^2}{1-\rho_{jk}^2} + \sum_{j \to k \in \mathcal{C}_T} \frac{\rho_{jk}^2}{\omega_{kk}}.$$

It is easy to see that $\theta_{jj}$ are identical in the two cases and this completes the proof. $\qquad\square$

## 5.2 Inverse Correlation Matrix Estimation

By Lemma 23, we can give an estimate of the inverse correlation matrix by the estimated CPDAG $\widehat{\mathcal{C}}_T$, sample correlations over the estimated tree skeleton, and estimated noise variance for each $j \in \widehat{V}_d$:

$$\hat{\theta}_{ij} = \begin{cases} -\hat{\rho}_{ij}/\hat{\omega}_{jj} & \text{if } i \to j \in \widehat{\mathcal{C}}_T \\ -\hat{\rho}_{ji}/\hat{\omega}_{ii} & \text{if } j \to i \in \widehat{\mathcal{C}}_T \\ -\hat{\rho}_{ij}/(1-\hat{\rho}_{ij}^2) & \text{if } i - j \in \widehat{\mathcal{C}}_T \\ \hat{\rho}_{ik}\hat{\rho}_{jk}/\hat{\omega}_{kk} & \text{if } i \to k \leftarrow j \in \widehat{\mathcal{C}}_T \\ 0 & \text{otherwise,} \end{cases} \quad \text{for } i \neq j \tag{13}$$

and

$$\hat{\theta}_{jj} = \begin{cases} \frac{1}{\hat{\omega}_{jj}} + \sum\limits_{j \to k \in \widehat{\mathcal{C}}_T} \frac{\hat{\rho}_{jk}^2}{\hat{\omega}_{kk}}, & j \in \widehat{V}_d, \\ 1 + \sum\limits_{j-k \in \widehat{\mathcal{C}}_T} \frac{\hat{\rho}_{jk}^2}{1-\hat{\rho}_{jk}^2} + \sum\limits_{j \to k \in \widehat{\mathcal{C}}_T} \frac{\hat{\rho}_{jk}^2}{\hat{\omega}_{kk}}, & j \in \widehat{V}_m. \end{cases} \tag{14}$$

Here, $\widehat{V}_d$ and $\widehat{V}_m$ are similarly defined through the estimated CPDAG $\widehat{\mathcal{C}}_T$ as in Lemma 22.

The CPDAG can be estimated through Algorithms 1 and 2. The sample correlations over the estimated skeleton can also be naturally defined. The remaining question is how to estimate noise variances in $\widehat{V}_d$. One natural method is to estimate $\omega_{jj}$ based on (5): $\hat{\omega}_{jj} = 1 - \sum\limits_{i \in \widehat{Pa}(j)} \hat{\rho}_{ij}^2$ for each $j \in \widehat{V}_d$, where $\widehat{Pa}(j)$ is the corresponding estimated parent set. However, the statistical property of this estimator is not easy to derive.

Instead, we propose to estimate the noise variance through the standard unbiased mean squared error, denoted as $\widehat{\omega}_{jj} = MSE_j$, of the least squares fit for the linear model

$$X_j = \sum_{i \in \widehat{Pa}(j)} \beta_{ij} X_i + \epsilon_j.$$

Under the Gaussian assumption, if $\widehat{\mathcal{C}}_T = \mathcal{C}_T$, we have $\widehat{V}_d = V_d$ and $\widehat{Pa}(j) = Pa(j)$ for each $j \in V_d$. Then, it is well-known that the least-squares MSE $\widehat{\omega}_{jj}$ is an unbiased estimate of $\omega_{jj}$. In fact, there holds

$$\widehat{\omega}_{jj} \overset{d}{=} \omega_{jj} \left( \frac{\chi^2_{n - d_j^{in}}}{n - d_j^{in}} \right), \quad \forall j \in V_d \tag{15}$$

where $d_j^{in}$ is the in-degree of node $j$, i.e. $d_j^{in} = |Pa(j)|$.

Finally, we introduce our result regarding the estimation error bounds of inverse correlation matrix estimation defined above.

**Theorem 24.** *Consider the linear polytree SEM (2) associated with a polytree $T = (V, E)$, where all variables have unit variances. Denote the minimum noise variance as*

$$\omega_{\min} = \min\{\omega_{jj} : j \in V_d\} \wedge \min\{1 - \rho_{ij}^2 : i - j \in \mathcal{C}_T\}.$$

*Denote $\nu$ as the total number of v-structures. It is easy to verify that all of these concepts only depend on the CPDAG $\mathcal{C}_T$. We make the assumption that*

$$\rho_{\max} \leq 1 - \delta \quad and \quad \omega_{\min} \geq \delta$$

*for some constant $\delta > 0$.*

*Assume that the estimated CPDAG $\widehat{\mathcal{C}}_T$ is obtained by Algorithms 1 and 2 with threshold $\gamma_{crit} \sqrt{(\log p)/n}$. If we assume (7) in Theorem 14 holds, i.e.*

$$\gamma_{crit} > C \quad and \quad n > \widetilde{C} \left( \frac{\log p}{\rho_{\min}^4} \right),$$

*where $C$ is as defined in Lemma 9. Then, with probability at least $1 - p^{-2}$, the estimated inverse correlation matrix defined in (13) and (14) satisfies*

$$\|\widehat{\Theta} - \Theta\|_{\ell_1} \leq \widetilde{C} (p + \nu) \sqrt{\frac{\log p}{n}}. \tag{16}$$

*In the above, $\widetilde{C}$ represents some constant that only depends on $\delta$ and $\gamma_{crit}$, whose value changes from line to line.*

*Proof.* In the following, we use $\widetilde{C}$ to represent a constant that only depends on $\delta$ and $\gamma_{crit}$, whose value can change from line to line. On the other hand, $C$ represents some absolute constant whose value changes from line to line.

Given (7) in Theorem 14 holds, with probability at least $1 - p^{-3}$, we have $\|\widehat{\rho} - \rho\|_{\max} < C\sqrt{\frac{\log p}{n}}$, and the true CPDAG is exactly recovered by, i.e. $\widehat{\mathcal{C}}_T = \mathcal{C}_T$. Consequently, the estimated noise variances satisfy (15).

Denote the maximum in-degree as
$$d_* = \max\{d_j^{in} : j \in V_d\} \vee 1.$$

From Corollary 7, we have $\rho_{\min} < 1/\sqrt{d_*}$. Then the assumption $n > \widetilde{C}\left((\log p)/\rho_{\min}^4\right)$ implies $n > \widetilde{C}d_*^2 \log p$. Then, based on concentration inequalities for Chi-square random variables, (15) implies that with probability at least $1 - p^{-3}$,
$$\max_{j \in V_d} |\widehat{\omega}_{jj} - \omega_{jj}| \leq C\sqrt{\frac{\log p}{n}}$$

for some absolute constant $C$.

With the above concentrations of $\widehat{\omega}_{jj}$'s and $\widehat{\rho}_{ij}$'s, based on the formula (13) and the assumption $\omega_{\min} \geq \delta$, we have for any $i \neq j$,
$$\begin{cases} |\hat{\theta}_{ij} - \theta_{ij}| \leq \widetilde{C}\sqrt{\frac{\log p}{n}} & \text{if } (i,j) \in \mathcal{T} \text{ or } i \to k \leftarrow j \in \mathcal{C}_T \text{ for some } k \\ |\hat{\theta}_{ij} - \theta_{ij}| = 0 & \text{otherwise}, \end{cases} \tag{17}$$

which further implies
$$\sum_{i \neq j} |\hat{\theta}_{ij} - \theta_{ij}| \leq \widetilde{C}\,(p + \nu)\,\sqrt{\frac{\log p}{n}}.$$

Further, (14) implies, for each $j = 1, \ldots, p$, there holds
$$|\widehat{\theta}_{jj} - \theta_{jj}| \leq \widetilde{C}(1 + d_j)\sqrt{\frac{\log p}{n}},$$

where $d_j$ is the degree of node $j$ in the skeleton $\mathcal{T}$. This implies
$$\sum_{j=1}^{p} |\hat{\theta}_{jj} - \theta_{jj}| \leq \widetilde{C}p\sqrt{\frac{\log p}{n}}.$$

Putting the above results together, we get (16). $\qquad\square$

# 6 Extension to Group Polytree Linear Structural Equation Models

In this section, we consider an extension of the linear polytree structural equation model (1) to the case of variable groups. It will become clear that this is an natural and straightforward extension of our theory, where substituting correlation with a multivariate counterpart to establishes the correlation decay property needed to proving the sufficient sample size for the correct graph recovery. Such group polytree or DAG models we describe below may arise when certain variables are closely related or driven by common latent variables. Assume the random vector $\boldsymbol{x} = [X_1, \ldots, X_p]^\top$ is partitioned into $p$ groups:
$$\boldsymbol{x}^\top = [\boldsymbol{x}_1^\top, \ldots, \boldsymbol{x}_p^\top],$$

where $\boldsymbol{x}_i$ is a $l_i$-dimensional random vector. We consider the following group linear polytree structural equation model over $T = (V, E)$:
$$X_j = \sum_{i=1}^{p} \boldsymbol{B}_{ij}^\top X_i + \boldsymbol{\epsilon}_j = \sum_{i \in Pa(j)} \boldsymbol{B}_{ij}^\top X_i + \boldsymbol{\epsilon}_j, \quad \text{for } j = 1, \ldots, p, \tag{18}$$

where the $l_i \times l_j$ matrix $\boldsymbol{B}_{ij} \neq \boldsymbol{0}$ if and only if $i \to j \in E$. Also, assume all $\boldsymbol{\epsilon}_j$'s are independent multivariate normal random vectors. If we denote
$$\boldsymbol{B} = \begin{bmatrix} \boldsymbol{0} & \boldsymbol{B}_{12} & \ldots & \boldsymbol{B}_{1p} \\ \boldsymbol{B}_{21} & \boldsymbol{0} & \ldots & \boldsymbol{B}_{2p} \\ \vdots & \vdots & \ddots & \vdots \\ \boldsymbol{B}_{p1} & \boldsymbol{B}_{p2} & \ldots & \boldsymbol{0} \end{bmatrix}$$

$\boldsymbol{\epsilon}^\top = [\boldsymbol{\epsilon}_1^\top, \ldots, \boldsymbol{\epsilon}_p^\top]$. Then the SEM can still be represented as $\boldsymbol{x} = \boldsymbol{B}^\top \boldsymbol{x} + \boldsymbol{\epsilon}$. Denote the covariance matrices

$$\mathrm{Cov}(\boldsymbol{x}) = \boldsymbol{\Sigma} = \begin{bmatrix} \boldsymbol{\Sigma}_{11} & \boldsymbol{\Sigma}_{12} & \ldots & \boldsymbol{\Sigma}_{1p} \\ \boldsymbol{\Sigma}_{21} & \boldsymbol{\Sigma}_{22} & \ldots & \boldsymbol{\Sigma}_{2p} \\ \vdots & \vdots & \ddots & \vdots \\ \boldsymbol{\Sigma}_{p1} & \boldsymbol{\Sigma}_{p2} & \ldots & \boldsymbol{\Sigma}_{pp} \end{bmatrix} \quad \text{and} \quad \mathrm{Cov}(\boldsymbol{\epsilon}) = \boldsymbol{\Omega} = \begin{bmatrix} \boldsymbol{\Omega}_{11} & \boldsymbol{0} & \ldots & \boldsymbol{0} \\ \boldsymbol{0} & \boldsymbol{\Omega}_{22} & \ldots & \boldsymbol{0} \\ \vdots & \vdots & \ddots & \vdots \\ \boldsymbol{0} & \boldsymbol{0} & \ldots & \boldsymbol{\Omega}_{pp} \end{bmatrix},$$

we still have $\boldsymbol{\Sigma} = (\boldsymbol{I} - \boldsymbol{B})^{-\top} \boldsymbol{\Omega} (\boldsymbol{I} - \boldsymbol{B})^{-1}$. Again, our goal is still to recover the polytree structural $T = (V, E)$ from $n$ i.i.d observation of $\boldsymbol{x}$: $\boldsymbol{x}_1, \ldots, \boldsymbol{x}_n$.

Our algorithm for group polytree learning is similar to that introduced in Section 2, with the pairwise sample correlations $\hat{\rho}_{ij}$ replaced with the leading sample canonical correlations between $\boldsymbol{X}_i$ and $\boldsymbol{X}_j$. To be specific, denote $\boldsymbol{R}_{ij} = \boldsymbol{\Sigma}_{ii}^{-\frac{1}{2}} \boldsymbol{\Sigma}_{ij} \boldsymbol{\Sigma}_{jj}^{-\frac{1}{2}} \in \mathbb{R}^{l_i \times l_j}$, then $\rho_{ij} \coloneqq \|\boldsymbol{R}_{ij}\|$ is the leading population canonical correlation coefficient between $\boldsymbol{X}_i$ and $\boldsymbol{X}_j$. Correspondingly, denote the sample version $\widehat{\boldsymbol{R}}_{ij} = \widehat{\boldsymbol{\Sigma}}_{ii}^{-\frac{1}{2}} \widehat{\boldsymbol{\Sigma}}_{ij} \widehat{\boldsymbol{\Sigma}}_{jj}^{-\frac{1}{2}}$, with the leading sample canonical correlation coefficient $\hat{\rho}_{ij} \coloneqq \|\widehat{\boldsymbol{R}}_{ij}\|$. Then we apply Algorithm 1 to recover the polytree skeleton and Algorithm 2 to recover the CPDAG, where $\hat{\rho}_{ij}$'s represent leading sample canonical correlations.

In analogy, we also denote $\eta_{ij} = \sigma_{\min}(\boldsymbol{R}_{ij})$ as pairwise least population canonical correlation coefficients. These quantities will not be employed in the algorithms, but will be used in our theoretical result of CPDAG recovery.

In the sequel, we aim to extend the consistency results Theorems 11 and 14 to the case of group polytree SEM. Since the sample canonical correlations are linear invariant, in theory, we can assume $\boldsymbol{X}_i \sim \mathcal{N}(\boldsymbol{0}, \boldsymbol{I}_{l_i})$ for each $i = 1, \ldots, p$ without loss of generality. In fact, if we replace $\boldsymbol{x}_i$ with $\widetilde{\boldsymbol{X}}_i = \boldsymbol{\Sigma}_{ii}^{-\frac{1}{2}} \boldsymbol{X}_i$, then

$$\widetilde{\boldsymbol{X}}_j = \sum_{i \in Pa(j)} \left( \boldsymbol{\Sigma}_{jj}^{-\frac{1}{2}} \boldsymbol{B}_{ij}^\top \boldsymbol{\Sigma}_{ii}^{\frac{1}{2}} \right) \widetilde{\boldsymbol{X}}_i + \boldsymbol{\Sigma}_{jj}^{-\frac{1}{2}} \boldsymbol{\epsilon}_j, \quad \text{for } j = 1, \ldots, p,$$

which shares the same polytree structure.

Before presenting our consistency results, we also need two lemmas. The first lemma is an extension of the correlation decay property, Lemma 3.

**Lemma 25.** *Consider the Gaussian group linear polytree model* (18) *with the associated polytree $T = (V, E)$. For each pair $(i, j)$, if the path connecting $i$ and $j$ is not a trek, we have $\rho_{ij} = 0$; if the path connecting $i$ and $j$ is a trek, we have*

$$\prod_{(s,t) \in \tau_{ij}} \eta_{st} \le \rho_{ij} \le \prod_{(s,t) \in \tau_{ij}} \rho_{st}.$$

*Proof.* Since population canonical correlations are linearly invariant, we can assume $\boldsymbol{X}_i \sim \mathcal{N}(\boldsymbol{0}, \boldsymbol{I}_{l_i})$ for each $i = 1, \ldots, p$ WLOG. Then, for each $i \to j \in E$, one can easily obtain

$$\boldsymbol{R}_{ij} = \boldsymbol{B}_{ij}, \quad \boldsymbol{R}_{ji} = \boldsymbol{B}_{ij}^\top.$$

Further, one can easily use argument of induction to show that for each pair $(i, j)$, if the path connecting $i$ and $j$ is not a trek, we have $\boldsymbol{R}_{ij} = \boldsymbol{0}$ and thereby $\rho_{ij} = 0$; if the path connecting $i$ and $j$ is a trek, denoted as

$$\tau_{ij} : i = v_l^L \leftarrow v_{l-1}^L \leftarrow \cdots \leftarrow v_1^L \leftarrow v_0 \to v_1^R \to \cdots \to v_{r-1}^R \to v_r^R = j,$$

we have

$$\boldsymbol{R}_{ij} = \boldsymbol{B}_{v_{l-1}^L v_l^L}^\top \boldsymbol{B}_{v_{l-2}^L v_{l-1}^L}^\top \cdots \boldsymbol{B}_{v_0 v_1^L}^\top \boldsymbol{B}_{v_0 v_1^R} \cdots \boldsymbol{B}_{v_{r-2}^R v_{r-1}^R} \boldsymbol{B}_{v_{r-1}^R v_r^R},$$

which further implies

$$\boldsymbol{R}_{ij} = \boldsymbol{R}_{v_l^L v_{l-1}^L} \boldsymbol{R}_{v_{l-1}^L v_{l-2}^L} \cdots \boldsymbol{R}_{v_1^L v_0} \boldsymbol{R}_{v_0 v_1^R} \cdots \boldsymbol{R}_{v_{r-2}^R v_{r-1}^R} \boldsymbol{R}_{v_{r-1}^R v_r^R}.$$

The fact $\rho_{ij} = \|\boldsymbol{R}_{ij}\|$ implies $\rho_{ij} \le \prod_{(s,t) \in \tau_{ij}} \rho_{st}$. $\qquad\square$

Another lemma is an extension of Lemma 9:

**Lemma 26.** *Consider a Gaussian group linear SEM* (18) *with $n \geq C_0(l_{\max} + \log p)$ for some absolute constant $C_0$, where $l_{\max} = \max_{1 \leq i \leq p} l_i$. Then, on an event $\mathcal{E}$ with probability at least $1 - 1/p^3$, the following inequality holds for some absolute constant $C$:*

$$\max_{1 \leq i < j \leq p} |\hat{\rho}_{ij} - \rho_{ij}| < C\sqrt{\frac{l_{max} + \log p}{n}}, \tag{19}$$

*where $\hat{\rho}_{ij}$ and $\rho_{ij}$ are sample and population leading canonical correlation coefficients between $\boldsymbol{X}_i$ and $\boldsymbol{X}_j$, respectively.*

*Proof.* Again, since population and sample canonical correlations are linearly invariant, we can assume $\boldsymbol{X}_i \sim \mathcal{N}(\boldsymbol{0}, \boldsymbol{I}_{l_i})$ for each $i = 1, \ldots, p$ WLOG. By the Remark 5.40 in Vershynin (2012) (reproduced in Appendix A), with $n \geq C_0(l_{\max} + \log p)$ for some sufficiently large constant $C_0$, one can show that with probability at least $1 - 1/p^3$, we have

$$\max_{1 \leq i \leq p} \|\widehat{\boldsymbol{\Sigma}}_{ii} - \boldsymbol{I}_{l_i}\| < C\sqrt{\frac{l_{max} + \log p}{n}}$$

and

$$\max_{1 \leq i < j \leq p} \|\widehat{\boldsymbol{\Sigma}}_{ij} - \boldsymbol{\Sigma}_{ij}\| < C\sqrt{\frac{l_{max} + \log p}{n}},$$

which further implies

$$\max_{1 \leq i < j \leq p} \|\widehat{\boldsymbol{R}}_{ij} - \boldsymbol{R}_{ij}\| < C\sqrt{\frac{l_{max} + \log p}{n}}.$$

Similar arguments can also be found in Ma & Li (2020). Then (19) follows from the fact

$$|\hat{\rho}_{ij} - \rho_{ij}| = |\|\widehat{\boldsymbol{R}}_{ij}\| - \|\boldsymbol{R}_{ij}\|| \leq \|\widehat{\boldsymbol{R}}_{ij} - \boldsymbol{R}_{ij}\|.$$

$\square$

With the above lemmas, the consistency of the recovery of polytree skeleton and CPDAG by Algorithms 1 and 2 with sample correlations replaced with sample leading canonical correlation coefficients can be straightforwardly established.

**Theorem 27.** *Consider a Gaussian group linear SEM* (18) *associated to a polytree $T = (V, E)$. Denote the minimum population leading canonical correlation coefficient, maximum population leading canonical correlation coefficient, and minimum population least canonical correlation coefficient over the tree skeleton as*

$$\rho_{\min} := \min_{i \rightarrow j \in E} |\rho_{ij}|, \quad \rho_{\max} := \max_{i \rightarrow j \in E} |\rho_{ij}|, \quad and \quad \eta_{\min} := \min_{i \rightarrow j \in E} |\eta_{ij}|.$$

*Assume $\rho_{\max} < 1 - \delta$ for some constant $\delta$. Denote by $\widehat{\mathcal{T}}$ the estimated skeleton by the Chow-Liu algorithm (Algorithm 1), and by $\mathcal{T}$ the true polytree skeleton. Then, with probability at least $1 - 1/p^3$, we have exact polytree skeleton recovery $\widehat{\mathcal{T}} = \mathcal{T}$ as long as*

$$n > C_0(\delta) \left( \frac{l_{\max} + \log p}{\rho_{\min}^2} \right)$$

*where $C_0(\delta)$ is a constant only depending on $\delta$.*

*Further, denote by $\mathcal{C}_T$ the true polytree CPDAG, and by $\widehat{\mathcal{C}}_T$ the estimated CPDAG from Algorithm 2 with threshold $\gamma_{crit}\sqrt{(l_{\max} + \log p)/n}$. If $\eta_{\min} > 0$, then, with probability at least $1 - 1/p^3$, we have $\widehat{\mathcal{C}}_T = \mathcal{C}_T$ as long as*

$$\gamma_{crit} > C \quad and \quad n > C_0(\delta) \gamma_{crit}^2 \left( \frac{l_{\max} + \log p}{\eta_{\min}^4} \right),$$

*where $C$ is an absolute constant, and $C_0(\delta)$ is a constant only depending on $\delta$.*

*Proof.* The proof is exactly the same as those of Theorems 11 and 14. Note that the sample size condition for CPDAG recovery relies on the minimum least population canonical correlation coefficient. In fact, if the ground truth is $i \leftarrow k \leftarrow j$ or $i \leftarrow k \rightarrow j$ or $i \rightarrow k \rightarrow j$, Lemma 25 only implies that $|\rho_{ij}| \geq \eta_{\min}^2$. $\square$

**Remark 28.** *Our CPDAG recovery result for group polytree SEM relies on the assumption $\eta_{\min} > 0$. At this moment, we don't know whether this is a necessary condition, and we plan to investigate this in future work.*

## 7 Numerical Experiments

To illustrate the feasibility and quantitative performance of the polytree learning method based on the Chow-Liu algorithm, we implement Algorithms 1 and 2 in Python and test on simulated data (Section 7.1). We further test on commonly used benchmark datasets (Section 7.2) to assess the robustness and applicability to real-world data. In all experiments, we set the threshold $\rho_{crit}$ (Algorithm 2) for rejecting a pair of nodes being independent based on the testing zero correlation for Gaussian distributions. Specifically, $\rho_{crit} = \sqrt{1 - \frac{1}{1 + t_{\alpha/2}^2/(n-2)}}$, where $t_{\alpha/2}$ is the $1 - \alpha/2$ quantile of a t-distribution with $df = n - 2$, and we use $\alpha = 0.1$. For comparisons, we run these same data using two basic and representative structural learning methods: the score-based hill climbing (Gámez et al., 2011), and the constraint-based PC algorithm (Spirtes et al., 2000) along with its early-stopping adaptation to polytree (Algorithm 3). We use R implementations of the hill climbing and the PC algorithm from `bnlearn` and `pcalg` packages, respectively, along with all the default options and parameters. We implemented the polytree-adapted PC algorithm in Python. An $\alpha = 0.01$ is used for the PC algorithm as recommended in Kalisch & Bühlman (2007). All codes are available at `https://github.com/huyu00/linear-polytree-SEM`. As is the case with all SEMs, caution should be taken when interpreting algorithm results in practical applications, as they represent potential causal interactions rather than definitive proofs.

We assess the results by comparing the true and inferred CPDAGs $\mathcal{C}$ and $\widehat{\mathcal{C}}$. On the skeleton level, there can be edges in $\mathcal{C}$ that are *missing* in $\widehat{\mathcal{C}}$, and vice versa $\widehat{\mathcal{C}}$ can have *extra* edges. For the CPDAG, we consider a directed edge to be *correct* if it occurs with the same direction in both CPDAGs. For an undirected edge, it needs to be undirected in both CPDAGs to be considered correct. Any other edges that occur in both CPDAGs are considered to have *wrong directions*. With these notions, we can calculate the False Discovery Rate (FDR) for the skeleton as $\frac{|\text{extra}|}{|\widehat{\mathcal{C}}|}$, and for the CPDAG as $\frac{|\text{extra}| + |\text{wrong direction}|}{|\widehat{\mathcal{C}}|}$. Here $|\text{extra}|$ is the number of extra edges, $|\widehat{\mathcal{C}}|$ is the number of edges in $\widehat{\mathcal{C}}$, and so on. To quantify the overall similarity and take into account the true positives, we calculate the Jaccard index (JI), which is $\frac{|\text{correct}| + |\text{wrong direction}|}{|\mathcal{C} \cup \widehat{\mathcal{C}}|} = \frac{|\text{correct}| + |\text{wrong direction}|}{|\text{missing}| + |\widehat{\mathcal{C}}|}$ for the skeleton, and $\frac{|\text{correct}|}{|\mathcal{C}| + |\widehat{\mathcal{C}}| - |\text{correct}|}$ for the CPDAG.

### 7.1 Testing on Simulated Polytree Data

Here we briefly describe how we generate linear polytree SEMs. Additional implementation details can be found in Section 7.3. First, we generate a polytree by randomly assigning directions to a random undirected tree. Next, the standardized SEM parameters $\beta_{ij}$'s are randomly chosen within a range, which in turn determine $\omega_{ii}$ (Eq. (5)). Motivated by the theoretical results (Theorems 11, 14 and 16), we make sure that in the above procedures, the generated SEM satisfies $\rho_{\min} \leq |\beta_{ij}| \leq \rho_{\max}$, the maximum in-degree $d_* = d_*^0$, and $\omega_{\min} \leq \omega_{ii}$ for all $i \in V$. Here $\rho_{\min}, \rho_{\max}, d_*^0, \omega_{\min}$ are pre-specified constants (values used are listed in Fig. 1 caption).

Figures 1 and 2 show the performance for $p = 100$ and $n$ ranging from 50 to 1000. We see that the Chow-Liu algorithm performs much better than hill climbing, and overall has an accuracy similar to or better than that of PC and early-stopping PC. At small sample sizes of less than 400, PC and early-stopping PC have a smaller FDR for skeleton recovery than Chow-Liu, but this is likely at the expense of the true positive rate, as reflected by the similar or lower JI of PC compared to Chow-Liu (Panels BD of Figs. 1 and 2). At larger sample sizes, Chow-Liu has a better accuracy in recovering the CPDAG. The early-stopping PC has a similar accuracy in recovering the skeleton as the regular PC and a better accuracy in recovering the

CPDAG (Panels CD of Figs. 1 and 2). This is likely due to the algorithm only applying Meek's Rule 1 (which is all needed for polytree) to orient the edges (Section 2.3.2). As $\rho_{\min}$ becomes smaller or as $d_*$ increases, the accuracy of Chow-Liu decreases, which is consistent with the theory (Theorems 11, 14 and 16). For hill climbing and PC, the accuracy is less affected by $\rho_{\min}$ or $d_*$ (Fig. 1 vs Fig. 2).

When comparing skeleton recovery (Panels A, B in Figs. 1 and 2) with CPDAG recovery (panels C, D in Figs. 1 and 2), the accuracy of skeleton recovery is higher, which is consistent with our theoretical results (Theorems 11, 14 and 16).

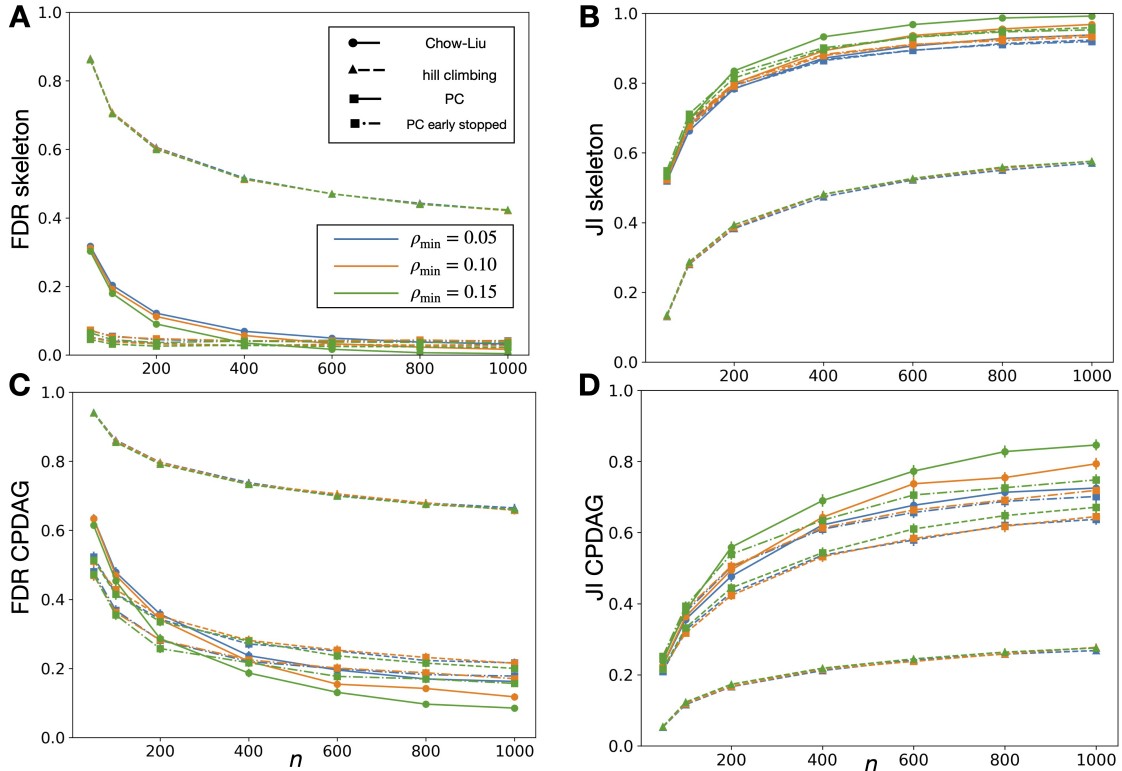

Figure 1: Performance on the polytree simulated data at $p = 100$ and the maximum in-degree $d_* = 10$. The results from the algorithms are represented by solid lines and dot markers (polytree), dash lines and triangle markers (hill climbing), solid lines and square markers (PC), and dash-dot lines and square markers (PC early stopped). Colors correspond to three different values of $\rho_{\min}$. The rest of the SEM parameters are $\rho_{\max} = 0.8$, and $\omega_{\min} = 0.1$. Panels A,C show the FDR (the smaller the better) for skeleton and CPDAG recovery. Panels B,D show the Jaccard Index (the larger the better). For each combination of SEM parameters, we randomly generate a polytree, the detailed generation of the $\beta_{ij}$'s and $\omega_{ii}$'s are described in Section 7.3. Then we draw iid samples from the SEM of different sizes (the x-axis, $n = 50, 100, 200, 400, 600, 800, 1000$). This entire process is repeated 100 times. Each point on the curves shows the average over the 100 repeats and the error bars are 1.96 times the standard error of the mean (many are smaller than the marker).

### 7.1.1 Running Time Comparison

Interestingly, the running time of the PC algorithm is significantly affected by $d_*$: the running time increases 40 folds when $d_*$ changes from 10 to 20 (Table 1), and the code may even fail to stop (running for more than 8 hours) when $d_* = 40$ (data not shown). This phenomenon can be explained by the relationship between the maximal number of neighbors and the maximal number of iterations in the PC algorithm; see Proposition 1 of Kalisch & Bühlman (2007). On the other hand, Chow-Liu is significantly more favorable in terms of running time, similar to the early-stopped PC which avoids the issue of maximal number of neighbors above. The Chow-Liu algorithm is up to 80 times faster than the slowest alternative algorithm

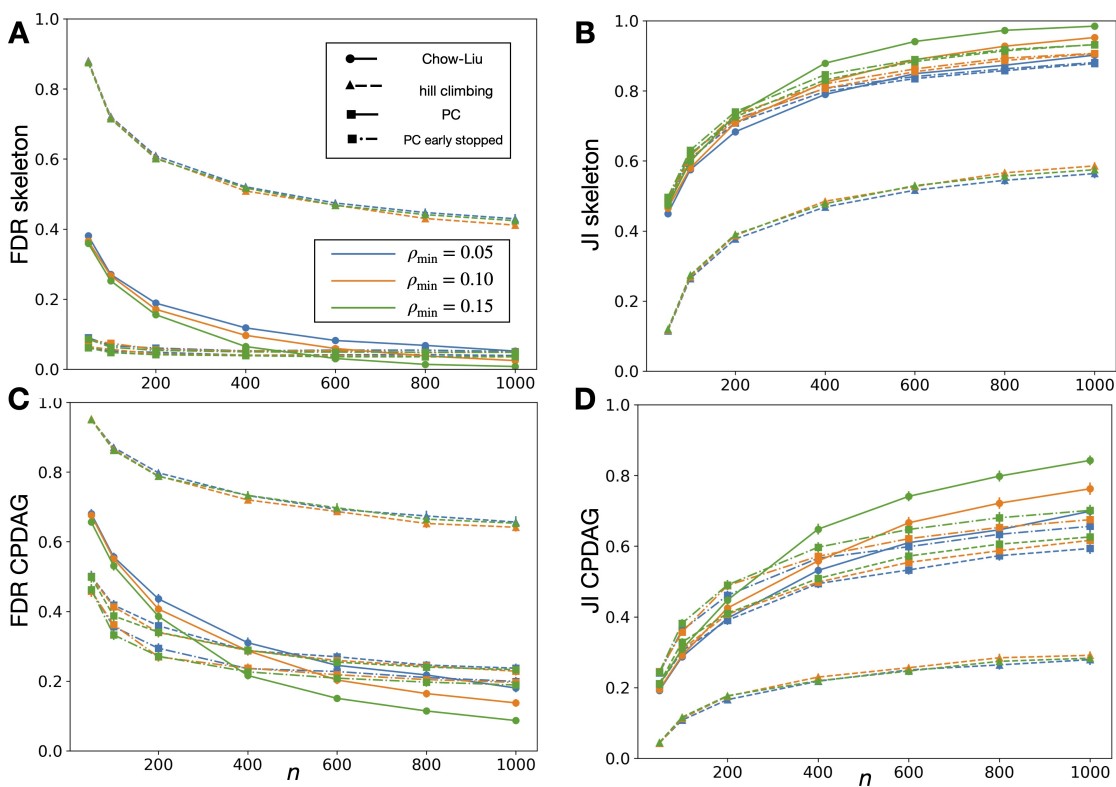

Figure 2: Same as Fig. 1 but for a maximum in-degree of $d_* = 20$.

(Table 1) and, importantly, has a running time that is constant across the SEM parameters (this is also true for all other experiments described later).

| (Unit: sec) | Polytree $p = 100$, $d_{\max}^{in} = 10$ | Polytree $p = 100$, $d_{\max}^{in} = 20$ | ASIA $p = 8$ | ALARM $p = 37$ |
|---|---|---|---|---|
| Chow-Liu | **0.01** | **0.01** | **0.0003** | **0.01** |
| Hill climbing | 0.87 | 1.00 | 0.012 | 1.48 |
| PC | 0.07 | 2.86 | 0.03 | 0.53 |
| PC early stopped | 0.03 | 0.03 | 0.001 | 0.38 |

Table 1: Running time comparison. The columns correspond to the SEM data in Figs. 1 and 2 (polytree), Table 2 (ALARM) and Table 3 (ASIA). $n = 5000$ for the AISA and ALARM. The running time is for one inference (averaged across trials/bootstraps when applicable). All computation is done on a 2019 Intel i7 quad-core CPU desktop computer.

## 7.2 Testing on DAG Benchmark Data

The ALARM dataset (Beinlich et al., 1989) is a widely used benchmark data. The true DAG (Fig. 3) is not a polytree and has 37 nodes and 46 edges. In fact, a three-phase algorithm initialized by Chow-Liu has been demonstrated to be effective on this data (Cheng et al., 2002). We simply conducted the Chow-Liu algorithm (Algorithms 1 and 2), and found that it still performs better than hill climbing and PC (including the early-stopping version) in terms of the metrics (Table 2) as well as intuitively by the inferred graph (Fig. 3). At $n = 5000$, it even achieves the best possible accuracy for skeleton recovery as Chow-Liu can achieve (there has to be at least $46 - (37 - 1) = 10$ edges missing in an inferred polytree).

Another benchmark we test is the ASIA dataset (Lauritzen & Spiegelhalter, 1988), which is a simulated DAG dataset with eight nodes. Note that the ground truth is sparse but not exactly a polytree. At $n = 500$

| $n = 500$ | Correct | Wrong d. | Missing | Extra | FDR sk. | JI sk. | FDR CPDAG | JI CPDAG |
|---|---|---|---|---|---|---|---|---|
| Chow-Liu | **28** | **4** | 14 | **4** | **0.11** | **0.64** | **0.22** | **0.52** |
| Hill climbing | 24 | 17 | **5** | 60 | 0.59 | 0.39 | 0.76 | 0.2 |
| PC | 14 | 17 | 15 | 13 | 0.3 | 0.53 | 0.68 | 0.18 |
| PC early stopped | 24 | 8 | 14 | 20 | 0.38 | 0.48 | 0.54 | 0.32 |
| $n = 5000$ | Correct | Wrong d. | Missing | Extra | FDR sk. | JI sk. | FDR CPDAG | JI CPDAG |
| Chow-Liu | 25 | 11 | 10 | **0** | **0.0** | **0.78** | **0.31** | 0.44 |
| Hill climbing | 27 | 18 | **1** | 62 | 0.58 | 0.42 | 0.75 | 0.21 |
| PC | 24 | 17 | 5 | 12 | 0.23 | 0.71 | 0.55 | 0.32 |
| PC early stopped | **42** | **1** | 3 | 39 | 0.48 | 0.51 | 0.49 | **0.49** |

Table 2: Performance on ALARM data. See text for the details of the accuracy measures: the number of correct, missing, extra, and wrong direction edges, FDR, and Jaccard index for skeleton and CPDAG. The best results across the algorithms are in bold.

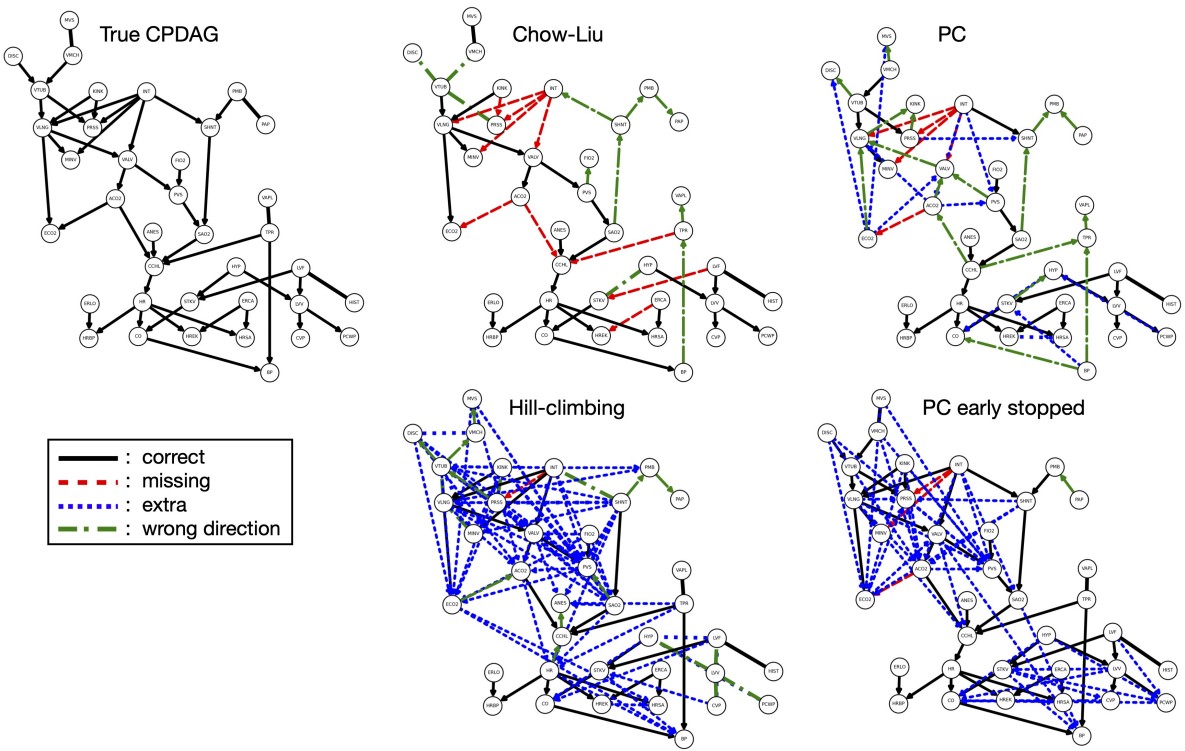

Figure 3: Comparing the true CPDAG of the ALARM data and the inferred one from the four algorithms at $n = 5000$. There are 37 nodes and 46 edges in the true CPDAG.

samples, the performance of Chow-Liu is comparable to that of hill climbing and PC algorithm, while the hill climbing gives the best result at $n = 5000$ (Table 3). We illustrate the comparison intuitively by plotting the most likely inference outcome of each algorithm across the bootstrap trials in Fig. 4 (where we re-sample $n$ observations from the original 5000 samples). Note the graph inferred by Chow-Liu (occurs at 23%) is the best possible result it can achieve. This is because at least one edge must be missing as the output is a polytree, and the v-structure involving B, E, and D can no longer be identified once missing the edge ED, leading to BD being undirected.

Lastly, we study a benchmark simulated dataset, EARTHQUAKE (Korb & Nicholson, 2010), whose ground truth graph is a polytree (Fig. 5). At $n = 500$ samples, the performance of Chow-Liu is comparable to that of hill climbing and PC algorithm, while Chow-Liu performs the best in the overall recovery of the skeleton and the CPDAG at $n = 2000$ (FDR and Jaccard index in Table 4). Similar to previous data, we plot the most likely inference outcome for each algorithm across trials Fig. 5. At $n = 2000$, the Chow-Liu algorithm perfectly recovers the true DAG in 90% of trials.

| $n = 500$ | Correct | Wrong d. | Missing | Extra |
|---|---|---|---|---|
| Chow-Liu | 4.0(1.1) | 1.8(1.25) | 2.2(0.6) | 1.2(0.6) |
| Hill climbing | **4.2(1.54)** | 2.3(1.35) | **1.5(0.67)** | 1.5(1.57) |
| PC | 3.3(1.19) | **1.7(0.9)** | 3.0(0.63) | **0.1(0.3)** |
| PC early stopped | 2.9(1.3) | 2.1(1.14) | 3.0(0.63) | 0.1(0.3) |
| $n = 5000$ | Correct | Wrong d. | Missing | Extra |
| Chow-Liu | 4.19(1.49) | 2.3(1.47) | 1.51(0.51) | 0.51(0.51) |
| Hill climbing | **6.96(0.9)** | **0.34(0.69)** | 0.7(0.57) | 0.87(0.93) |
| PC | 4.59(1.06) | 1.79(1.04) | 1.62(0.56) | **0.15(0.39)** |
| PC early stopped | 3.82(0.83) | 3.53(1.09) | **0.65(0.54)** | 1.05(0.57) |

(continue)

| $n = 500$ | FDR sk. | JI sk. | FDR CPDAG | JI CPDAG |
|---|---|---|---|---|
| Chow-Liu | 0.17(0.09) | 0.64(0.11) | 0.43(0.16) | 0.38(0.14) |
| Hill climbing | 0.17(0.14) | **0.7(0.14)** | 0.46(0.22) | **0.39(0.23)** |
| PC | **0.02(0.06)** | 0.62(0.09) | **0.36(0.23)** | 0.35(0.14) |
| PC early stopped | 0.02(0.06) | 0.62(0.09) | 0.44(0.25) | 0.3(0.16) |
| $n = 5000$ | FDR sk. | JI sk. | FDR CPDAG | JI CPDAG |
| Chow-Liu | 0.07(0.07) | 0.77(0.11) | 0.4(0.21) | 0.41(0.19) |
| Hill climbing | 0.1(0.1) | **0.83(0.11)** | **0.13(0.15)** | **0.78(0.18)** |
| PC | **0.02(0.05)** | 0.78(0.08) | 0.29(0.17) | 0.48(0.14) |
| PC early stopped | 0.12(0.06) | 0.82(0.08) | 0.54(0.11) | 0.31(0.09) |

Table 3: Performance on ASIA data. The accuracy measures (the number of correct, missing, extra, and wrong direction edges, FDR and Jaccard index for skeleton and CPDAG; see text) are averaged over 1000 bootstraps (resampling $n$ observations out of 100,000) and the standard deviations are in the parentheses. The best results across the algorithms are in bold.

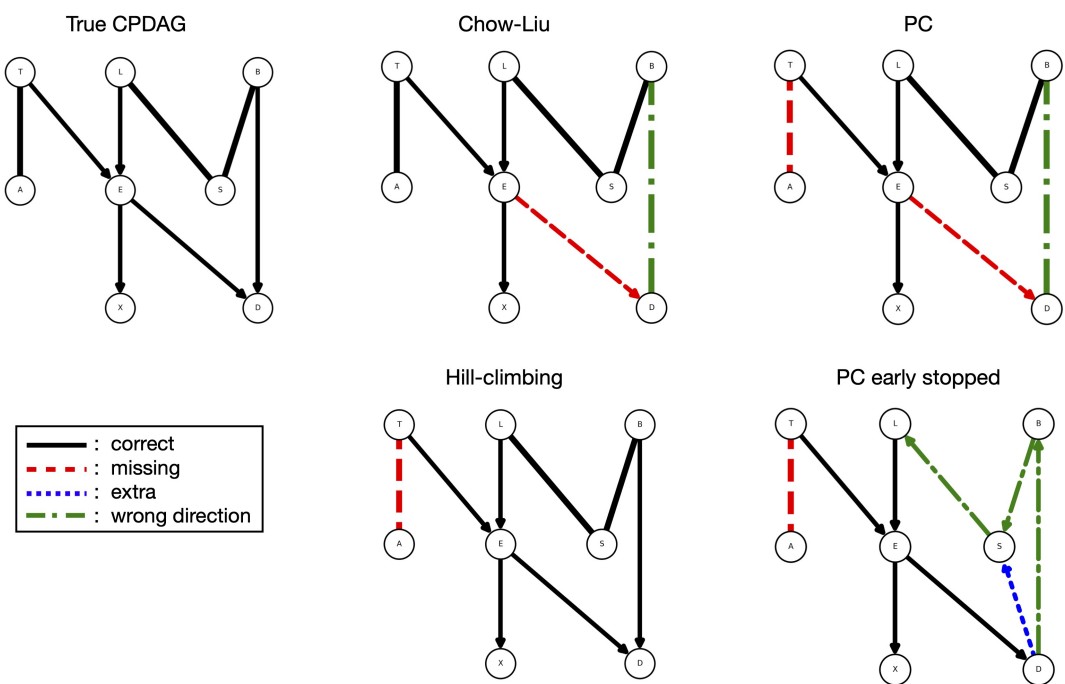

Figure 4: The true CPDAG and the typical inferred CPDAG for the ASIA data with $n = 5000$ samples. We plot the most likely inferred graph across 1000 bootstraps for each algorithm, which occurs at 23% (Chow-Liu), 44% (hill climbing), 42% (PC), 50% (early-stopping PC), respectively.

## 7.3 Details on polytree data generation

In simulated polytree data, we draw i.i.d. samples from a Gaussian linear SEM with a polytree structure. First, we generate an undirected tree with $p$ nodes from a random Prufer sequence. The Prufer sequence which has a one-to-one correspondence to all the trees with $p$ nodes is obtained by sampling $p - 2$ numbers with replacement from $\{1, 2, \ldots, p\}$. Next, a polytree is obtained by randomly orienting the edges of the

| $n = 500$ | Correct | Wrong d. | Missing | Extra |
|---|---|---|---|---|
| Chow-Liu | 2.87(1.55) | 0.83(1.27) | 0.3(0.67) | **0.3(0.67)** |
| Hill climbing | **3.38(1.29)** | **0.46(1.07)** | **0.17(0.48)** | 0.85(0.75) |
| PC | 2.52(1.45) | 1.16(1.13) | 0.32(0.56) | 0.66(0.64) |
| PC early stopped | 2.63(1.53) | 1.09(1.22) | 0.28(0.57) | 0.76(0.74) |
| $n = 2000$ | Correct | Wrong d. | Missing | Extra |
| Chow-Liu | 3.62(1.17) | 0.38(1.16) | 0.01(0.08) | **0.01(0.08)** |
| Hill climbing | **3.88(0.64)** | **0.12(0.64)** | **0.0(0.0)** | 0.61(0.61) |
| PC | 3.86(0.47) | 0.14(0.47) | 0.0(0.0) | 0.62(0.61) |
| PC early stopped | 3.68(0.78) | 0.32(0.78) | 0.0(0.0) | 0.79(0.79) |

| (continue) | | | | |
|---|---|---|---|---|
| $n = 500$ | FDR sk. | JI sk. | FDR CPDAG | JI CPDAG |
| Chow-Liu | **0.08(0.17)** | **0.89(0.22)** | 0.28(0.39) | 0.68(0.4) |
| Hill climbing | 0.17(0.14) | 0.81(0.17) | **0.26(0.3)** | **0.72(0.31)** |
| PC | 0.14(0.13) | 0.81(0.18) | 0.43(0.33) | 0.51(0.35) |
| PC early stopped | 0.15(0.14) | 0.8(0.18) | 0.42(0.35) | 0.54(0.36) |
| $n = 2000$ | FDR sk. | JI sk. | FDR CPDAG | JI CPDAG |
| Chow-Liu | **0.0(0.02)** | **1.0(0.04)** | **0.08(0.27)** | **0.91(0.27)** |
| Hill climbing | 0.13(0.11) | 0.87(0.11) | 0.16(0.19) | 0.84(0.19) |
| PC | 0.13(0.11) | 0.87(0.11) | 0.16(0.14) | 0.84(0.16) |
| PC early stopped | 0.17(0.14) | 0.83(0.14) | 0.24(0.24) | 0.74(0.26) |

Table 4: Performance on EARTHQUAKE data. The accuracy measures (the number of correct, missing, extra, and wrong direction edges, FDR and Jaccard index for skeleton and CPDAG; see text) are averaged over 1000 bootstraps (resampling $n$ observations from a total of 100,000 samples) and the standard deviations are in the parentheses. The best results across the four algorithms are in bold.

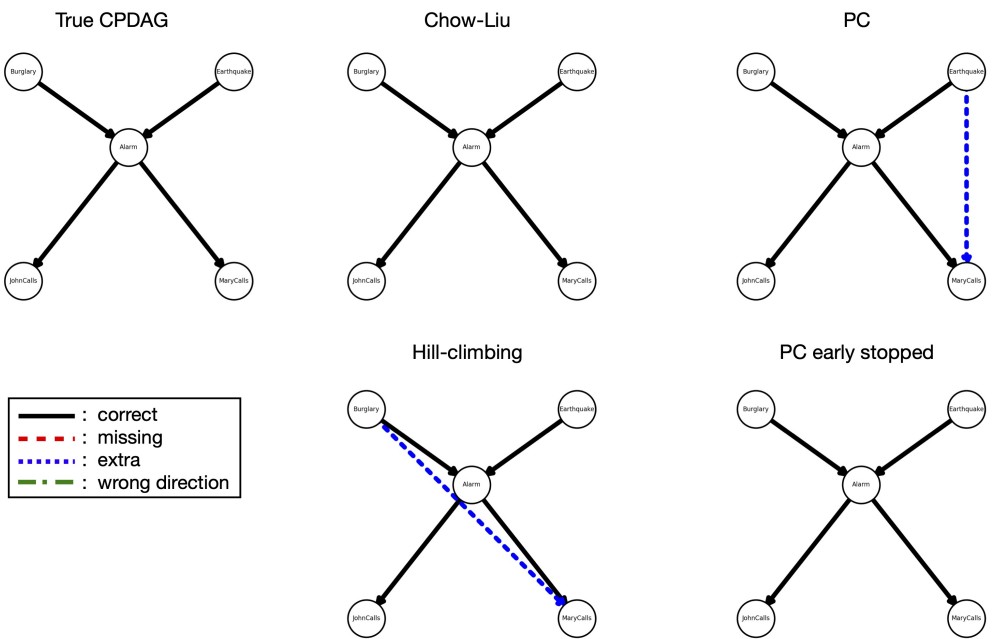

Figure 5: The true CPDAG and the most frequently inferred CPDAG for the EARTHQUAKE data with $n = 2000$ samples over 1000 trials. The graph shown occurs at 90% for Chow-Liu, 47% for hill climbing, 46% for PC, and 41% for early-stopping PC, respectively.

undirected tree. We also ensure that one of the nodes has a specified large in-degree $d_*^0$. This is done by making a node $i$ occur at least $d_*^0 - 1$ times in the Prufer sequence, so the node will have a degree at least $d_*^0$ in the undirected tree. We then orient all edges connected to $i$ by selecting $d_*^0$ of them to be incoming edges. The rest of the edges in the tree are oriented randomly as before.

In the next step, we choose the value of the standardized $\beta_{ij}$ corresponding to the correlations. Note that once $\beta_{ij}$'s are given, $\omega_{ii}$ are determined by Eq. (5). Motivated by the theoretical conditions on $n, p$ such as those in Theorems 11 and 14, we choose $\beta_{ij}$ according to some pre-specified values $\rho_{\min}$ and $\rho_{\max}$ and study the effects of these parameters on the recovery accuracy. To avoid ill-conditioned cases, we require that $\omega_{ii} \geq \omega_{\min}$, where $\omega_{\min}$ is another parameter. This adds constraints on $\beta_{ij}$, $\sum_{j=1}^{p} \beta_{ij}^2 \leq 1 - \omega_{\min}$, in addition to $\rho_{\min} \leq |\beta_{ij}| \leq \rho_{\max}$. We sample $\beta_{ij}^2$ uniformly among the set of non-negative values satisfying the above inequality constraints. This sampling is implemented by drawing $\beta_{ij}^2$, (corresponding to all the edges in the polytree) sequentially in a random order as $\min(\rho_{\max}^2, \rho_{\min}^2 + v_j x)$, where $x$ is drawn from the beta distribution $B(1, \tilde{d}_j^{\text{in}})$. Here $\tilde{d}_j^{\text{in}}$ is the number of incoming edges to node $j$ whose $\beta_{ij}^2$ has not yet been chosen, and $v_j = 1 - \omega_{\min} - d_j^{\text{in}} \rho_{\min}^2 - \sum_k \beta_{kj}^2$, where the sum is over all edges $k \to j$ whose $\beta_{kj}^2$ have already been chosen, $d_j^{\text{in}}$ is the total number of incoming edges to $j$. The use of beta distribution here is based on the fact of the order statistics of independent uniformly distributed random variables. As an exception, we first set two $|\beta_{ij}|$ values to attain equality in the constraints by $\rho_{\min}$ and $\rho_{\max}$ before choosing the rest of $\beta_{ij}$'s according to the above sampling procedure. For $\rho_{\max}$, we randomly choose a node $i$ that satisfies $\rho_{\min}^2(d_i^{\text{in}} - 1) + \rho_{\max}^2 \leq 1 - \omega_{\min}$, $d_i^{\text{in}} > 0$ (always exists if $\rho_{\max}^2 + \omega_{\min} \leq 1$ and the minimum nonzero in-degree is 1), and set one of its incoming edges to have $|\beta_{ji}| = \rho_{\max}$. For $\rho_{\min}$, we choose a node among the rest of nodes with $d_k^{\text{in}} > 0$ and set $|\beta_{lk}| = \rho_{\min}$ for one of its incoming edges. Lastly, a positive or negative sign is given to each $\beta_{ij}$ with equal probability. After the $\beta_{ij}$'s (i.e., matrix $\boldsymbol{B}$) are chosen (and hence $\boldsymbol{\Omega}$), the samples $\boldsymbol{x}_1, \ldots, \boldsymbol{x}_n$ are drawn according to $\boldsymbol{x} = (\boldsymbol{I} - \boldsymbol{B})^{-\top} \boldsymbol{\epsilon}$, where $\boldsymbol{\epsilon}$ are zero mean Gaussian variables with covariance $\boldsymbol{\Omega}$.

# 8 Discussion

This paper studies the problem of polytree learning, a special case of DAG learning where the skeleton of the directed graph is a tree. This model has been widely used in the literature for both prediction and structure learning. We consider the linear polytree model, and consider the Chow-Liu algorithm (Chow & Liu, 1968) that has been proposed in Rebane & Pearl (1987) for polytree learning. Our major contribution in this theoretical paper is to study the sample size conditions under which the polytree learning algorithm recovers the skeleton and the CPDAG exactly. Under certain mild assumptions on the correlation coefficients over the polytree skeleton, we show that the skeleton can be exactly recovered with high probability if the sample size satisfies $n > O((\log p)/\rho_{\min}^2)$, and the CPDAG of the polytree can be exactly recovered with high probability if the sample size satisfies $n > O((\log p)/\rho_{\min}^4)$. We also establish necessary conditions on sample size for both skeleton and CPDAG recovery, which are consistent with the sufficient conditions and thereby give a sharp characterization of the difficulties for these two tasks. In addition, we also study inverse correlation matrix estimation under the linear polytree SEM. Under the component-wise $\ell_1$ metric, we give an estimation error bound that is characterized by the dimension, the sample size, and the total number of v-structures.

There are a number of remaining questions to study in the future. It would be interesting to study how to relax the polytree assumption. In fact, the benchmark data analysis (Section 7.2) is very insightful, since it shows that the considered Chow-Liu based CPDAG recovery algorithm, which seemingly relies heavily on the polytree assumption, could lead to reasonable and accurate structure learning result when the ground truth deviates from a polytree to some degree. This inspires us to consider the robustness of the proposed approach against such structural assumptions. For example, if the ground truth can only be approximated by a polytree, can the structure learning method described in Sections 2.3.1 and 2.3.2 lead to an approximate recovery of the ground truth CPDAG with theoretical guarantees? Similarly, if the sample size is not large enough and the CPDAG is thereby unable to be recovered exactly, can we still obtain an accurate estimate of the inverse correlation matrix? As aforementioned, polytree modeling is usually used in practice only as initialization, and post-processing could give better structural recovery results. A well-known method of this type is given in Cheng et al. (2002) without theoretical guarantees. An interesting future research direction is to include such post-processing steps into our theoretical analysis, such that our structural learning results (e.g., Theorems 14) hold for more general sparse DAGs.

## Acknowledgment

X. Li and X. Lou acknowledge support from the NSF via the Career Award DMS-1848575. Y. Hu was partly supported by an Early Career Scheme grant 26303921 from the Hong Kong Research Grants Council. We would like to thank J. Peng for helpful discussions.

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

## A    Preliminaries

**Lemma 1 in Kalisch & Bühlman (2007)** *Let $\hat{\rho}_{ij}$ and $\rho_{ij}$ are sample and population correlation between $X_i$ and $X_j$ as in Section 2. Consider the Gaussian linear polytree SEM* (1) *with* $\sup_{n, i \neq j} \rho_{ij} < M < 1$. *Then for any* $0 < \gamma \leq 2$, *the following inequality holds:*

$$\sup_{i \neq j} \mathbb{P}\left(|\hat{\rho}_{ij} - \rho_{ij}| \geq \gamma\right) \leq C_1 (n - 2) \exp\left((n - 4) \log\left(\frac{4 - \gamma^2}{4 + \gamma^2}\right)\right),$$

*where* $0 < C_1 < \infty$ *depends only on* $M$.

**Remark 5.40 in Vershynin (2012)** *Assume that $A$ is an $N \times n$ matrix whose rows $A_i$ are independent sub-gaussian random vectors in $\mathbb{R}^n$ with second moment matrix $\Sigma$. Then for every $t \geq 0$, the following inequality holds with probability at least $1 - 2\exp(-ct^2)$:*

$$\frac{1}{N}\|A^* A - \Sigma\| \leq \max(\delta, \delta^2), \quad \text{where } \delta = C\sqrt{\frac{n}{N}} + \frac{t}{\sqrt{N}}.$$

*Here $C = C_K$ and $c = c_K > 0$ depend only on the sub-gaussian norm $K = \max_i \|A_i\|_{\psi_2}$ of the rows.*

