# OpenReview forum: "Learning Linear Polytree Structural Equation Model"
_TMLR — Accepted by TMLR_

### Review · Reviewer_xRcG · 2024-11-28

**Summary Of Contributions:**

This paper studies the structure learning of directed polytree with linear Gaussian SEM. The main focus is on analyzing the well-known Chow-Liu algorithm for learning polytrees. To this end, some other well-known properties of linear Gaussian SEMs are leveraged for establishing
- sufficient sample complexity of learning the skeleton (Theorem 11) and CPDAG (essentially, Markov equivalence class) (Theorem 14),  and extension of these results to group polytree linear SEMs (Section 6)
- lower bounds for the structure learning (Theorem 16),
- an adaptation of the PC algorithm to linear Gaussian polytrees (Theorem 19).
- specializing the known precision matrix estimation results to polytrees (Theorem 24).

The proofs are rather straightforward and follow by specializing the existing results to linear Gaussian polytrees setting. That being said, to my knowledge, no other paper has taken this initiative. By investigating several aspects of the problem (e.g., relating the necessary and sufficient sample sizes, adapting the PC algorithm etc.), the paper does a good job of analyzing and cleaning up the analysis of Chow-Liu algorithm for linear Gaussian polytrees.

**Audience:**

Yes

**Claims And Evidence:**

Yes

**Requested Changes:**

- I’d recommend authors expand the related work discussion to include structure learning of linear SEMs without polytree restriction. Some relevant references can be Peters and Bühlmann (2014) and Ghoshal and Honorio (2018), though I’m sure there are more recent relevant results as well. A brief discussion of these results, with a possible comparison with the results in the polytree setting, can be useful to position the paper and highlight the refinement achieved by assuming a polytree structure.

- I believe the authors aimed the paper to be mostly self-contained. To achieve this, I think some of the existing results that are crucial for the results in this paper can be given in the appendix. For instance, the results of Vershynin (2012) and Kalish and Bühlmann (2007) are repeatedly leveraged in the proofs. Explicitly stating those results in the appendix can make the paper more self-contained. Similarly, the supposedly trivial omitted proofs (e.g. proof of Lemma 23) can be added to the appendix.

- Correlation decay: This is overly emphasized in the introduction, but is not discussed in detail in the main sections. I feel like there can be a better discussion on its role and why it’s necessary (using references and/or examples).

- Section 6: Some explanation on the motivation and importance of this extension would have been nice. I am not aware of much work in this line (group polytree linear SEMs), so it may be worth it to explain *why* it’s an extension that’s worthy of attention in this paper.

- On experiments, it might be worth it to further explore non-polytrees via synthetic graphs, e.g., what happens when the graph to be learned is not exactly a polytree but differs from the nearest polytree by only a few edges.

The following points are mostly minor nitpicks. I simply suggest the authors consider the following points when revising the paper, but they are not crucial for the acceptance of the paper.

- Page 3: Before eq.(3), it is said that $\epsilon_j$’s are usually with different variances. In principle, I agree with it. However, despite being a limitation, structure learning literature commonly assumes that noise variances are equal (e.g., Peters and Bühlmann (2014)).

- Theorem 11: Isn’t $C$ used here already defined in Lemma 9? So instead of Lemma, it should be Lemma 9 in Theorem 11 statement.

- Section 5 intro: Perhaps this can be more upfront and mention the subsequent results. I think the importance of this section is the nice connection between precision matrix estimation and the graph via v-structures. Considering $\Theta_{ij}$ in eq. (12), this is not surprising. Nevertheless, to my knowledge Theorem 24 is novel, and the appearance of v-structures in the estimation guarantees can be mentioned at the beginning of the section to prepare the reader.

- $\gamma_{\rm crit}$. Does $crit$ refer to critical? It appears a little bit out of context when first introduced on Page 5. It says that the choice of threshold is discussed in subsequent sections, but perhaps it warrants one more sentence to discuss its role on Page 5. Also, see typo in $\rho_{cirt}$, and inconsistency of $\gamma_{\rm crit}$ vs. $\gamma_{crit}$. Also, I’m not sure whether using *both* $\rho_{crit}$ and $\gamma_{crit}$ is intentional.
- Page 6: $\mathbf{B}$ is permutationally similar to a *strictly* upper triangular matrix.
- Remark 28: missing period at the end of sentence.
- Section 7.2: Perhaps mention earlier that ALARM is not a polytree.

Peters, J., & Bühlmann, P. (2014). Identifiability of Gaussian structural equation models with equal error variances. Biometrika, 101(1), 219-228.

Ghoshal, A., & Honorio, J. (2018). Learning linear structural equation models in polynomial time and sample complexity. In International Conference on Artificial Intelligence and Statistics (pp. 1466-1475).

**Strengths And Weaknesses:**

Following up on my comments from the previous paragraph: The paper is well-written with great clarity and clean flow. The results are not surprising, the proofs mostly build on existing results and do not contain much novelty — which can be considered a weakness. However, not every paper needs to be very novel, especially for TMLR. I believe this paper can be a very helpful reference for the overall community as it ties up many aspects of the linear Gaussian polytree structure learning problem -- from sufficient and necessary sample sizes, distinctions between skeleton recovery and v-structure recovery, and adaptation of the PC algorithm.

---

> ### Author Response · Authors · 2025-01-10
> **Replies to Review by Reviewer xRcG**
>
> **Re: Requested Changes**
>
> * Discussion of linear SEMs without polytree restriction: We added a short paragraph at the bottom of page 2 including the two suggested citations. Since our study is largely motivated by the Chow-Liu algorithm, which is specific to the polytree, we refrained from an extended discussion of the general linear SEM literature but referred the readers to see the discussions in Ghoshal and Honorio 2018.
> * Self-contained: The results from Vershynin (2012) and Kalish and Bühlmann (2007)  are now included in the Appendix and referred to. We have included the proof of Lemma 23 (in the main text since it is short).
> * Correlation decay: In our opinion, correlation decay is the crux to understanding the proof of the sufficient sample size and that is why we included a detailed discussion in the introduction, including its use in undirected graphs and the classic results of Rebane & Pearl. We have now added explicit reference to the correlation decay property (beginning of Section 3; proof of Theorem 11, 14, Lemma 17; before Lemma 25), and elaborated how we obtained the case of group polytree in Section 6 as a natural extension based on the idea of correlation decay.
> * Section 6 motivation: The extension to the group polytree is mainly motivated as a natural multivariate extension of our theory, where the key machinery of correlation decay can be analogously established when substituting correlation with the largest canonical correlations.  From the application perspective, variable group polytrees may arise when certain variables are closely related or driven by common latent variables. We have added some explanations as these in the beginning of Section 6.
> * Synthetic non-polytrees experiments: We agree that this is indeed an interesting and important topic. However, given the focus of the paper being on theory and we do not yet have theoretical results on non-polytree cases, we think the experimental and theoretical studies of this topic are better left for future work.
>
> **Re: Minor requests:** We have revised and corrected accordingly. Below are some specific changes.
>
> Re: “Page 3: Before eq.(3), noise variances”: We have removed “usually with difference variances” before Eq.(2) to de-emphasize it.
>
> Re: “Section 5 intro”: Thank you for the suggestion. We have added in the introduction of Section 5: “It may be noteworthy that the error bound we obtained (Theorem 24) depends on the total number of v-structures in addition to the usual dimension and sample size.”
>
> Re: “$\gamma_{crit}$”: We have clarified the relationship of the two notations when first introduced on page 5 when $|\hat \rho_{ij}|< \rho_{crit} = \gamma_{crit} \sqrt{(\log p)/n}$, where the choice of threshold or critical value is discussed in subsequent sections.”

---

### Review · Reviewer_WnuG · 2024-12-25

**Summary Of Contributions:**

This paper presents an analysis of a classical algorithm to recover the causal structure from data under a particular case of the structural equation model (SEM) with polytree dependencies. The authors characterize the number of samples needed to recovering such structure successfully under different regimes. They also include other extensions such as the estimation of the inverse correlation matrix. Along with the theoretical study, they present numerical experiments.

**Audience:**

Yes

**Broader Impact Concerns:**

Practical applications of algorithms for detecting causal structure might pose ethical challenges. Even if this work is theoretical and does not delve into particular applications, a cautionary note about extracting causal meaning out of DAG recovery would be beneficial for practitioners that may use this work.

**Claims And Evidence:**

Yes

**Requested Changes:**

- Please add a description of the importance of studying DAG recovery under the polytree model.
- Please add experiments showing that the bounds predicted by the theory are realized in practice.

Editorial comments:
- In paragraph 1 of the introduction: “For a summary, see the survey papers Drton…” -> “For a summary, see the survey papers by Drton…” In general, revise the sentences where the citation is inlined (\citet) they should treat it as if they are talking about the authors and not about the paper.
- In paragraph 2 of the introduction, towards the end, the authors state “[these] models have received a significant amount of research interest” and proceed to cite to papers that are more than 20 years old. Adding newer references would be useful.
- The authors use the phrase “roughly speaking” twice in the introduction. I think that using an alternative like “in broad terms” would be more appropriate for a written format.
- In paragraph 5 of the introduction, I do not understand the sentence “it has been shown in Rebane & Pearl (1987) that there holds the “mutual information decay”…”
- At the beginning of Section 2.2, “Let’s” -> “Let us”
- Section 3, first paragraph: “the famous Wright’s formula” -> “Wright’s famous formula”
- Throughout section 3 and beyond, I found confusing that C is used as a scalar constant and also as a CPDAG. For example, C_0 in Lemma 9 and C_T in Theorem 14.
- In Theorem 11, do the authors mean that “C is defined in Lemma 18”? Should it not be Lemma 9?
- In the first paragraph of the proof of Theorem 11, “Corollary 3” -> “Lemma 3”
- In the title of section 8, I think it is more standard the use if the singular “Discussion”.

**Strengths And Weaknesses:**

**Strengths**

The authors clearly state that they are analyzing an existing classical algorithm. For this, they present novel theoretical results that help shed light on the algorithm’s behavior. They also  present experimental results showing that the algorithm is on par with more modern alternatives.

**Weaknesses**

I would like to see a paragraph describing why polytree models are meaningful, for example describing applications or examples where they arise. Saying that DAG models are popular seems a little bit too general.

My main concerns lies with the experimental results. I was expecting to see studies showing a phase transition when the number of samples reaches the theoretical prediction. If the bounds are sharp, we should be able to clearly see a qualitative change in the behavior of the algorithm when the number of samples is sufficient or not. The current experiments showing that the Chow-Lou algorithm is effective, but they fail to support the specific contributions of this paper.

---

> ### Author Response · Authors · 2025-01-10
> **Replies to Review by Reviewer WnuG**
>
> **Re: Polytree motivation:** We discussed the motivation for considering polytree models along with related literature in the second paragraph of the Introduction. The main reason is tractability, similar to the rationale for studying undirected tree models.
>
> **Re: Experiments showing theoretical bounds:** Our theory characterizes the sample size up to a constant, and the constants for the sufficient and necessary sample sizes could differ (and neither constant is optimal). This means our theory has not proven a phase transition per se. Establishing a phase transition result would require different techniques and analysis than the paper.
> On the other hand, our main result that recovering the CPDAG requires more samples than recovering the skeleton is demonstrated in our numerical experiments (Figures 1 and 2, comparing top vs bottom panels). We have added a discussion to emphasize this at the end of Section 7.1
>
> **Re: Editorial comments**
>
> We have revised/corrected the manuscript according to the comments. Below are some specific changes.
>
> Re: “In paragraph 2 of the introduction…”: We have added a citation to Chatterjee & Vidyasagar 2022 and Tramontano et al. 2022.
>
> Re: “In paragraph 5 of the introduction…”: Please see if the revised sentence, with an added reference to the equation within the citation, is clear: “it has been shown in Rebane & Pearl (1987) (see their Eq. 13) that there holds a ‘mutual information decay’ along the skeleton of the polytree.”
>
> Re: “Throughout section 3...”: Notation for CPDAG is now changed to $\mathcal{C}_T$ throughout to avoid confusion.
>
> **Re: Broder Impact Concerns:** We have added a cautionary note in the beginning of the Numerical Experiments Section 7: As is the case with all SEMs, caution should be taken when interpreting algorithm results in practical applications, as they represent potential causal interactions rather than definitive proofs.”

---

### Review · Reviewer_kpqu · 2024-12-27

**Summary Of Contributions:**

This paper investigates conditions on the sample size for polytree learning under the linear SEM model. It provides sample size conditions for recovering the skeleton and CPDAG using the Chow-Liu algorithm, where the conditions are related to the correlation bounds between variables and the estimation error for finite sample sizes. Additionally, it establishes sufficient conditions for the sample size and derives probability bounds for recovery failure when the sample size is insufficient. Furthermore, it examines the estimation error bounds of the inverse correlation matrix in the linear polytree SEM and validates the theoretical findings through numerical experiments.

**Audience:**

Yes

**Broader Impact Concerns:**

No concerns

**Claims And Evidence:**

Yes

**Requested Changes:**

(1) I think the structure of the theoretical section is very clear, but the experimental section is somewhat confusing. First, it would be better to considering adjusting the placement of the figures to make them closer to the corresponding text. It is a bit hard to follow when the text and the figure are separated by 5 pages (e.g., Figure 5).

(2) In the experimental section, it would be better to make the content intended for analysis more distinct and independent. For example, consider separating the running time comparison in Section 7.1 into its own subsection for more focused analysis.

(3) Please consider citing and discussing the following paper.

Tramontano D, Monod A, Drton M. Learning linear non-Gaussian polytree models, UAI 2022.

**Strengths And Weaknesses:**

Strengths:
1. The topic is important as it focuses on determining the necessary sample size to reliably recover the true underlying graph structure

2. It extends the convergence analysis from tree-structured undirected graphical models to directed polytrees, which enable the modeling of directional causal relationships. By leveraging the unique structural properties of polytrees, it bridges the gap between undirected dependency structures and directed causal inference frameworks.

3. It derives sample size bounds for both accurate recovery and failure probabilities in skeleton and CPDAG estimation. These bounds are sharp, offering valuable insights into the trade-off between sample complexity and estimation reliability.

Weaknesses:
The Chow-Liu algorithm, on which the proposed method is based, is more sensitive to data assumptions compared to other methods and exhibits a slower convergence rate.

---

> ### Author Response · Authors · 2025-01-10
> **Replies to Review by Reviewer kpqu**
>
> **Re: Weaknesses:**
> The polytree assumption is a limitation and it would be interesting to systematically study how robust the results are for non-polytree DAGs in future work. Our experiments on benchmark data which include two non-polytree cases do suggest that it could be promising (Fig. 3 and 4).
> We are not sure which section/result is being referred to regarding the sensitivity and slower convergence rate, so please elaborate if needed. We have shown that the Chow-Liu algorithm achieves the optimal sample size needed for exact recovery with matching sufficient and lower bound results up to a constant (Theorem 11, 14, and Theorem 16).
>
>
> **Re: Requested Changes:**
>
> (1): The figure and table placements have been improved to go closely to text.
>
> (2): A new subsection is created for running time comparison.
>
> (3): We have added a citation to the paper at the end of the second paragraph of the Introduction.

---

### Decision · Action_Editor_wR2S · 2025-02-25

**Recommendation:** Accept as is

**Comment:**

This paper examines the required sample size conditions for polytree learning within the linear SEM model. It establishes conditions for recovering the skeleton and CPDAG using the Chow-Liu algorithm, linking them to the correlation bounds between variables and the estimation error in finite sample scenarios. The theoretical finds are verified by the experiments. While the results in this paper are not very novel and surprising, all the reviewers believe this is a solid work. Based on the review comments, I recommend acceptance of this paper.

**Audience:**

Yes

**Claims And Evidence:**

Yes